# Remediation of Potential Toxic Elements from Wastes and Soils: Analysis and Energy Prospects

**Alberto González-Martínez [1], Miguel de Simón-Martín [1], Roberto López [2], Raquel Táboas-Fernández [1] and Antonio Bernardo-Sánchez [3,\*]**

[1]   Department Area of Electrical Engineering, School of Mining Engineering, University of León (Spain), Campus de Vegazana s/n, 24071 León, Spain; alberto.gonzalez@unileon.es (A.G.-M.); miguel.simon@unileon.es (M.d.S.-M.); raqueltaboas@gmail.com (R.T.-F.)

[2]   Department Area of Physical Chemistry, Faculty of Biological and Environmental Sciences, University of León (Spain), Campus de Vegazana, s/n, 24071 León, Spain; rlopg@unileon.es

[3]   Department Area of Mines Engineering, School of Mining Engineering, University of León (Spain), Campus de Vegazana s/n, 24071 León, Spain

\*   Correspondence: antonio.bernardo@unileon.es; Tel.: +34-987-29-35-54

**Abstract:** The aim of this study is to evaluate the application of the main hazardous waste management techniques in mining operations and in dumping sites being conscious of the inter-linkages and inter-compartment of the contaminated soils and sediments. For this purpose, a systematic review of the literature on the reduction or elimination of different potential toxic elements was carried out, focusing on As, Cd and Hg as main current contaminant agents. Selected techniques are feasible according to several European countries' directives, especially in Spain. In the case of arsenic, we verified that there exists a main line that is based on the use of iron minerals and its derivatives. It is important to determine its speciation since As (III) is more toxic and mobile than As (V). For cadmium (II), we observed a certain predominance of the use of biotic techniques, compared to a variety of others. Finally, in mercury case, treatments include a phytoremediation technique using *Limnocharis flava* and the use of a new natural adsorbent: a modified nanobiocomposite hydrogel. The use of biological treatments is increasingly being studied because they are environmentally friendly, efficient and highly viable in both process and energy terms. The study of techniques for the removal of potential toxic elements should be performed with a focus on the simultaneous removal of several metals, since in nature they do not appear in isolation. Moreover, we found that energy analysis constitutes a limiting factor in relation to the feasibility of these techniques.

**Keywords:** hazardous waste; contaminated soil; potential toxic elements; removal; mine waste

## 1. Introduction

Waste is the unusable result of a material after it has been used to develop a job or operation [1]. In the EU, total waste production amounted to 2.5 billion tons every year [2]. The increase in waste generation produced over last 50 years is the main reason for environmental legislation enacting. Actually, the main guideline in EU environmental policy is the 7th Environment Action Program, which promotes the protection, conservation, and enhancement of the natural capital Member States as one of the main objectives for 2020.

The transposition of EU directives on waste matter to Spanish legislation promoted the enactment of Law 22/2011, 28th July, on waste and contaminated soils. Based on this set of new laws, waste can be defined as any substance or object that its holder discards or intends to dispose of [3].

Law 22/2011, 28th July, aims to "regulate the management of waste by promoting measures to prevent its generation and mitigate adverse impacts on human health and the environment associated

with its generation and management, improving efficiency in the use of resources and regulating the legal regime of contaminated soils". The importance of waste management lies in the fact that, if it is not done, the environment could deteriorate irreversibly.

A waste generation comparison of European countries between 2004 and 2014 is shown in Figure 1. The circle line represents the waste amount generated in 2004, while the solid circle represents the waste generation in 2014. In Spain, as in other European countries, waste generation has been linked to economic growth, so the trend of recent years has been to reduce the volume of generated waste (e.g., reduction factor of 0.69 in Spain and 0.49 in Portugal). However, some countries, as Latvia, Sweden, Netherland, and Norway, increased their waste generation to a factor of 2.09, 1.82, 1.43, and 1.42 in a decade [4]. EU annual report on waste statistics shows the main reasons of this increment to be the high shares of major mineral wastes (because their relatively sizeable mining and quarrying activities) and construction and demolition activities of that countries [5].

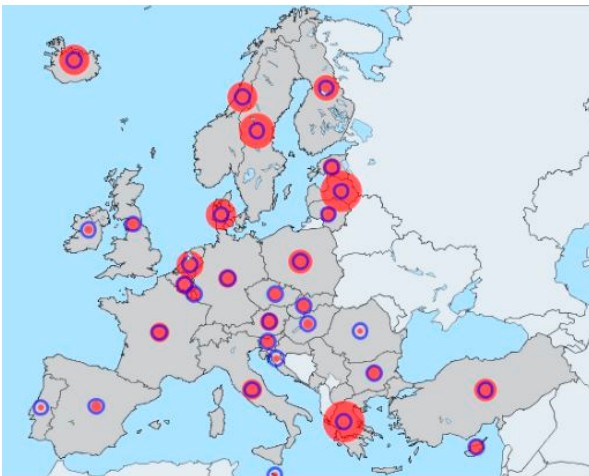

**Figure 1.** Waste generation comparison in EU Member States for the 2004 (circle line) to 2014 (solid circle) period. Source: [4].

The contribution of the different activities to the generation of waste in Spain in 2014 are presented in Figure 2. More than half of the waste generated in 2014 came from the services, construction and mining sectors. The waste generated in industry and in the services sector are different in nature, while in mining and construction waste is mostly of a mineral kind [6].

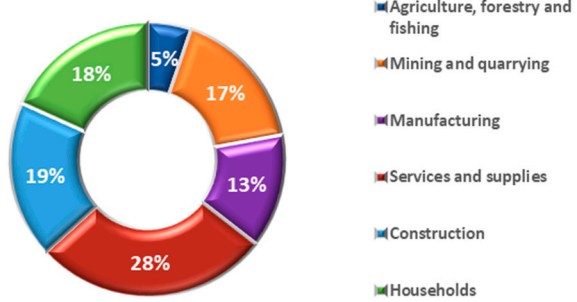

**Figure 2.** Generation of waste by economic activity in Spain in 2014. Adapted from: [6].

The extractive industry obtains minerals through mining techniques, which include drilling and blasting operations, among others. In addition, for commercial use, these extracted minerals must benefit from mineral processing techniques. In both stages of the process, waste is generated. In this sense, mining, or extractive industries wastes are the solid, aqueous, or paste residues that remain after

the investigation and use of a geological resource, provided that such material constitutes waste as it is defined in Law 22/2011, on waste and contaminated soils [3].

The Environmental European Agency defines percolation as a process which concerns the movement and filtering of fluids through porous materials [7]. In soil contamination, this term is associated with the movement of polluted water solutions across different land layers. These solutions, also called *leachates*, are characterized to displace large quantities of heavy metals, resulting in clean soils contamination. As a consequence, leaching is found to be one of the worst environment problems caused by mining wastes. These leachates cause serious environmental damage, triggering the contamination of large amounts of lands with dangerous metals at a long distance away from the contamination source. In Carnoulès (France), mining installations were abandoned 55 years ago. An acid stream with pH between 2.5–4.7 and 50–350 mg/L of arsenic flowed into the Amous river, with a very high environmental impact [8]. On the other hand, in Giant mine (Canada), 237,000 tons of arsenic trioxide dust is currently safe, but when the rock at Giant Mine was crushed and mined out, the arsenic was exposed to the environment. Large sections of the underground mine were backfilled with waste rock and tailings. The arsenic concentration in these sources was hundreds of times lower than in the arsenic trioxide waste in the storage chambers. However, the large volumes meant they also could contaminate the surrounding groundwater [9].

One of the most important problems in relation to potential toxic elements is that they are not biodegradable. [10]. As a consequence, heavy metals become toxic at very low concentrations. This toxicity depends on the concentration, the chemical form, and the persistence in which they occur, since in trace concentrations they may be indispensable for living beings (Na, K, Mg, Ca, V, Mn, Fe, Co, Ni, Cu, Zn, and Mo) [11].

Metals, as those cited before, coexist with other cations and anions in acid water solutions because their high insolubility in alkaline pH [12]. It is also true that acid solutions are frequently found in wastewater produced in industries such as mining, metal processing, electroplating, textiles, tanning, and oil refining, as well as in leaching from some dumpling sites. As a result, mining waste is characterized to produce severe environmental damage when an efficient treatment is not applied to these emissions.

Khalid et al. found mercury (Hg), lead (Pb), cadmium (Cd), and arsenic (As) to be the main metal(-oids) produced by metalliferous mining and smelting activities [13]. In addition, the World Health Organization identify the same metals/metalloids to be the major public health concerns [14]. As a consequence, these are the four metals/metalloids most widely studied in the literature. Between them, mercury, cadmium, and arsenic have been found to have high significant presence in mining waste products and dumping sites leaching [15]. This is the reason why this work is focused on the available disposal techniques for their remediation.

The choice of a suitable toxic elements' remediation technique considers the properties of both the elements and the contaminated site. Any treatment is associated with the modification of toxicity, mobility and volume of the waste, whether by physical, chemical, or biological processes. These treatments can be classified according to several factors. Classification according to the type of treatment may be described as follows [16]:

Biological treatments or bioremediation: Based on the use of the metabolic activity of organisms like plants, fungi, or bacteria for the transformation or elimination of the pollutants.

Physic-chemical treatments: Based on the use of the physical and/or chemical properties of the contaminants or the medium to remove, separate, or contain them.

Thermal treatments: Based on the use of heat to increase volatilization or melt contaminants.

The main advantages and disadvantages of these technologies are summarized in Table 1.

**Table 1.** Advantages and disadvantages of biological, physical-chemical, and thermal treatments for arsenic, cadmium, and mercury remediation. Adapted from [16] and [17].

| Treatment | Advantages | Disadvantages |
|---|---|---|
| Biological | Economical<br>Sustainable<br>Removes contaminants<br>Minimal or no further treatment needed | Longer treatment time<br>Need for toxicity verification<br>Only suitable if soil allows microbial growth<br>Very sensitive to process conditions changes |
| Physic-chemical | Economical (intermediate)<br>Short application periods<br>Easier to control | Need for waste generated treatment<br>Recovery systems for extraction fluids |
| Thermal | | Energy and engineering needs<br>Possible contaminant gaseous emissions<br>Expensive |

The remediation of environments contaminated with potential toxic elements by chemical methods can be excessively expensive due to the specificity of potential toxic elements reactivity. This specificity can make in-situ treatments to be more difficult to carry on when different metals must be removed from the area. On the other hand, biological methods can be applied both in-situ and ex-situ with high elimination specificity due to (1) the specific conditions in which metabolic organisms' reactions take place, and (2) the wide range of organisms with different metabolic routes suitable to soil bioremediates [18].

A controversy about scientific evidence for environmental damages caused by the use of bioremediation continues today. On one hand, Swannwell et al. reported some information about field evaluations of marine oil spill bioremediation [19]. In the *Exxon Valdez* incident description, they showed evidence that bioremediation causes less environmental damage than conventional techniques. However, several works indicated that microbes used in bioremediation may produce metabolites and other residues (i.e., dead biomass) which could make soils be even more toxic after bioremediation than they were before [20,21]. As a consequence, (1) high volume of solid waste should be managed at the end of the remediation process [22], and (2) soil characteristics must be studied wide enough before deciding to apply this remediation technique as many factors (degree of exposure, metal concentration, temperature, salinity) can completely change whether bioremediation is suitable or not to be used as a specific soil treatment. Furthermore, it has been a wide agreement on the interest in biological techniques, mainly characterized by its safety and low cost. In addition, they can be applied at the contaminated site. However, bioremediation processes produce residues [22]. Carrying out on-site treatment involves other advantages such as reducing exposure and avoiding problems associated with accidental discharges during transport. The increase in the use of biological techniques in relation to traditional physic-chemical techniques is due to the fact that the latter involve the alteration or elimination of the ecosystem and its physic-chemical properties, despite the fact that these technologies are faster and more effective [23,24].

This paper itself is innovative in terms of conducting a systematic review of the most common contaminant agents considering not only the remediation effects, but also analyzing the involved energy needs and prospects, which will make the different treatments more or less sustainable. It will represent the leading work for a more efficient and sustainable design of remediation techniques than can be applicable not only to hazardous wastes from mining or dumping sites, but also from environmental disasters or intensive potential toxic elements activities. The systematically conducted review will constitute a significant contribution and, from the author's point of view, a nexus between environmental researchers and energy researchers, which is new in the literature in this field.

Finally, the better the treatment techniques are known and optimized, the better locally available energy sources can be used to feed them, especially considering renewable energy sources. The authors observe that a high energy consumption of certain treatments can constitute a real barrier for their effective application as it can make them unaffordable due to the associated costs for the energy supply

and the impossibility of supplying the required energy just by using renewable energy sources, such as photovoltaic systems, with an average specific energy capacity of about 170 W/m$^2$. Thus, the local remediation techniques that can take advantage of locally generated renewable energy sources have become of high interest, reducing the overall lifecycle impact on the environment. Thus, this paper fits as an outstanding application of locally available energy sources and sustainability. The final target is to remediate the human impact on the environment without producing, or at least minimizing the production, of greenhouse gas emissions, reducing the dependence on imported fuels, and creating economic development and improving local employment, i.e., in a sustainable way.

### 1.1. Arsenic

Arsenic is a chemical element that belongs to the group of metalloids or semi-metals, due to its intermediate behavior between metals and non-metals. It can be found in four states of oxidation: +5 (arsenate), +3 (arsenide), 0 (arsenic) and −3 (arsine), often as metal sulphides, arsenides, or arsenates. In surface environments, the predominant forms are trivalent [As (III)] and pentavalent [As (V)] arsenic, which occur as oxyanions. There are also methylated derivatives of arsenic such as monomethylarsonic acid (MMAA), dimethylarsinic acid (DMAA), and trimethylarsine oxide (TMAO) [25]. In water, it is mostly presented as arsenate (+5), but under anaerobic conditions, it is presented as arsenide (+3).

The toxicity of arsenic depends on the state of oxidation in which it is found and/or the chemical compound of which it is part. The inorganic species are always more toxic than the organic ones. The most toxic form is arsine ($AsH_3$), followed by arsenic trioxide ($As_2O_3$), arsenide [As (III)], and arsenate [As (V)].

The main route of arsenic exposure is ingestion through water and food. Water pollution is a global problem, as the World Health Organization's recommended maximum concentration limit for drinking water (10 μg As/L) has been exceeded in many places [26].

The presence of arsenic in the environment can cause serious risks to health, such as neuropathy, conjunctivitis, diabetes, renal damage, an enlarged liver, bone marrow depression, high blood pressure, cardiovascular disease or skin, lung and liver cancer [15].

### 1.2. Cadmium

Cadmium is a metal found in the Earth's crust with an abundance in soil of 0.16 ppm. Its most common oxidation state is +2. Most of this element is produced as a by-product of zinc smelting, although it can also be produced as greenockite (CdS) [27]. It is a potential toxic element that causes environmental problems and can be accumulated in the human body through the food chain [28]. This accumulation may cause skeletal damage, evidenced by low bone mineralization. In addition, cadmium is also carcinogenic if inhaled [15].

Environmental contamination by cadmium occurs frequently in areas surrounding zinc-, lead- or copper-smelting factories [29]. Scenarios such as the incineration of waste and other materials containing cadmium as a pigment or stabilizer, the burning of fossil fuels, and the use of phosphate fertilizers can also be sources of soil or air pollution [30].

The World Health Organization recommends a maximum concentration limit of cadmium in water for human consumption of 3 μg Cd/L [26]. The US Agency for Toxic Substances and Disease Registry (ATSDR) lists cadmium as the sixth-most dangerous metal according to its prevalence and severity due to the intoxication it causes.

### 1.3. Mercury

In these substances, mercury is in an oxidation state of +2, +1 and smaller. It is usually found as cinnabar (HgS), which may contain drops of metallic mercury, metallic mercury, mercury vapor, inorganic mercury (I) and (II), alkylmercury, and phenylmercury [25].

Mercury is one of the most polluting elements for humans and many animals due to its high toxicity. The different species of mercury present different characteristics in terms of environmental behavior,

bioavailability (as rate of which a toxin is absorbed) [31], metabolism, and effects on organisms. It must be in ionic forms to show harmful effects. The World Health Organization defined mercury to be responsible for abnormal brain development, resulting in a lower IQ, and consequently a lower earning potential. In addition, mercury affects other aspects in brain and produces cardiovascular diseases [15]. Mercury salts exhibit high toxicity when dissolved in water. Alkylmercury has irreversible effects on the nervous system, and methylmercury has serious teratogenic, carcinogenic, and mutagenic effects [25]. In addition, mercury is easily absorbed by biota (especially in its organic forms), which leads to its accumulation in the food chain.

The maximum allowable limit of Hg (II) ions in drinking water is 1 μg Hg/L [32], and the maximum limit recommended for inorganic mercury, by the World Health Organization, is 6 μg Hg/L [26].

## 2. Systematic Review

A systematic review is defined as "an observational and retrospective research design, which synthesizes the results of multiple primary investigations" [33]. In this case, such a review has been carried out in order to assess the state of the art in the field of mining-waste management and the remediation of contaminated soils and leaches. The following conditions have been considered in the review:

Sources: Journal articles and technical review reports have been included, provided that they have been indexed and involved a peer-review process.
Language: English.
Period of publication: Research articles published between 2004 and 2017 have been included.
Requirements: All studies analyzed are freely accessible.
Keywords: "Removal mining", "arsenic", "cadmium", "mercury".
Type: Original research contributions and review works have been considered.

The systematic review was performed after the search conditions were established. 210 documents on arsenic removal, 190 on cadmium, and 137 on mercury were consulted; 537 documents were thus analyzed in total. A distribution of number or documents studied by year is shown in Figure 3.

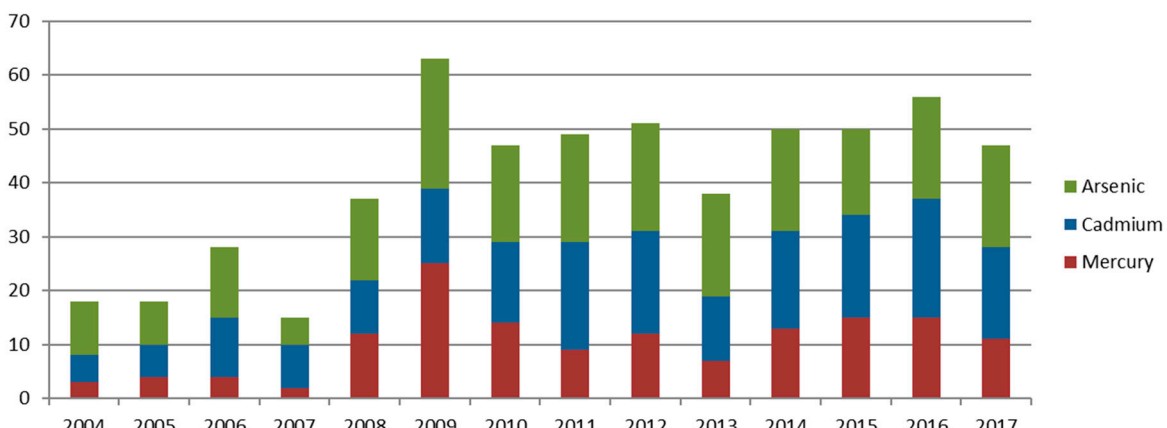

**Figure 3.** Number of references, organized by publication year, resulting from the systematic review of studies on potential toxic elements removal techniques (As, Cd, and Hg) used in the preparation of this article.

An initial search was conducted in bibliographic databases, applying the conditions established above. Through this process, Table 2 results were obtained. These results refer to general techniques for eliminating each of the potential toxic elements considered in this work: Arsenic, cadmium, and mercury (keywords).

**Table 2.** Number of references from the systematic review of potential toxic elements removal techniques.

| Metal/Metalloid | Articles | Reviews |
|---|---|---|
| Arsenic | 210 | 15 |
| Cadmium | 190 | 7 |
| Mercury | 137 | 9 |
| **Total** | **537** | **31** |

The previous search was filtered according to the specific elimination techniques analyzed, through which the results shown in Table 3 were obtained.

**Table 3.** Number of references from the systematic review of specific techniques for the removal of potential toxic elements (As, Cd, and Hg).

| Metal/Metalloid | Articles | Reviews |
|---|---|---|
| **Arsenic** | | |
| Adsorption | 48 | 4 |
| Electrocoagulation | 2 | - |
| Precipitation | 47 | 2 |
| Bioleaching | 3 | - |
| Others | 110 | 9 |
| **Cadmium** | | |
| Adsorption | 38 | 2 |
| Sequential soil washing | 1 | - |
| Biosynthesis of metal chalcogenides | 1 | - |
| Others | 150 | 5 |
| **Mercury** | | |
| Phytoremediation | 6 | 1 |
| Nanobiocomposite hydrogel | 3 | 1 |
| Others | 141 | 3 |
| **TOTAL** | **537** | **31** |

From the evaluated references, a 5.24%, 3.16%, and 2.19% of them face the problem of arsenic, cadmium, and mercury removal from the leachates, respectively. This sort of references has increased significantly in the recent years.

## 2.1. Arsenic

In relation to techniques for reducing arsenic, there is a predominant line based on the use of iron. Among these techniques, some of them should be highlighted: Co-precipitation and natural adsorption by iron minerals [34,35], electrocoagulation using iron electrodes [36], use of iron-rich sludge from coal treatment [37], sorption using coconut shell and iron oxide-coated sand [38], and biogenic ferric precipitates [39]. There are also biotic arsenic removal techniques, such as bioleaching using *Acidothiobacillus thiooxidans*.

### 2.1.1. Co-Precipitation and Natural Adsorption by Iron Minerals

Arsenic co-precipitation can be performed by natural attenuation, produced in leaching mine process, or by using a specific installation to develop it.

Natural attenuation is based on ferrihydrite and schwertmannite formation during acid mine drainage. Arsenic is incorporated into their chemical structure during their formation, causing a solid precipitate in aqueous solution. The natural attenuation of arsenic presents in acid mine effluents occurs principally through the co-precipitation of As (V) with schwertmannite. The main advantages of this method are its low cost and high efficiency, making arsenic concentrations in water negligible [34]. However, this process can only be produced in environments where ferrihydrite and schwertmannite are formed.

Ferrihydrite and schwertmannite formation can also produce goethite. Goethite has a high specific active surface [35], which adsorbs arsenic. This phenomenon is called natural adsorption because it is produced without the need of a designed installation.

At last, arsenic can be precipitated by using ferric salts in a designed reactor. Figure 4 shows a schematic model of a co-precipitation arsenic system.

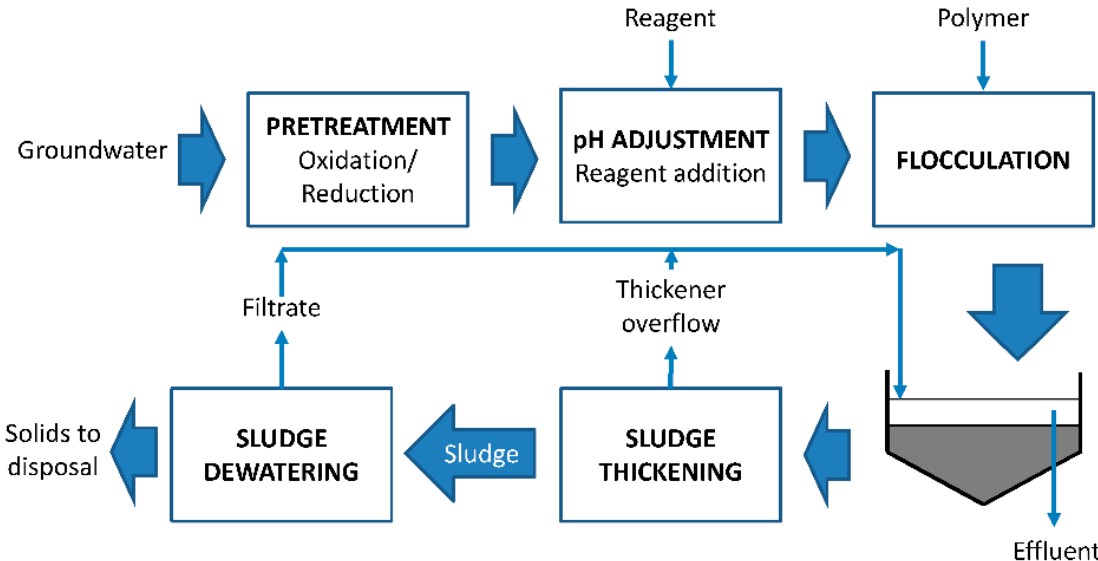

**Figure 4.** Schematic diagram a co-precipitation arsenic system. Adapted from: [40].

In a reagent-based system, ferric salts are added to residual water, making arsenic form a solid matrix in the aqueous solution. Afterwards, ferric-arsenic salts are separated from water and conducted to a sludge treatment. Solubility of As (V) is lower than As (III). As a consequence, sludge stability over time is also responsible for the oxidation of As (III) before neutralization. These are the main reasons why a pretreatment of As (III) oxidation is sometimes included.

### 2.1.2. Hydrometallurgical Process Based on Arsenic Precipitation with Lime

One of the most important technology for arsenic removal is neutralization with lime and the recirculation of precipitation sludge to enhance their properties (stability, water content). Arsenic is precipitated as calcium arsenite and/or calcium arsenate at pH 11–12. The ratio of Fe/As and the proportion of As (III)/As (V) are important parameters that influence the performances of As removal and sludge stability [36].

As a global method, lime neutralization is a relatively economic process for arsenic immobilization. However, the precipitates show poor long-term stability and therefore must be deposited in hazardous landfills. [36].

### 2.1.3. Electrocoagulation Using Iron Electrodes

Electrocoagulation with iron electrodes is a promising technique in comparison to those which need reagents addition because the limitation of oxidizers and absorbents reaction and absorption capabilities.

Both As (III) and As (V) are present in arsenic contaminated groundwater. When electric current passes through iron electrodes, Fe (II) dissolves. In the presence of dissolved oxygen, Fe (II) oxidizes to Fe (III) and forms a solid matrix, $FeAsO_4$ and $Fe(OH)_3$ onto which As can be sorbed. Thus, arsenic is removed from the aqueous phase by separating the precipitates. The optimum pH for As (III) removal is 7, using low intensity, in absence of other substances. In addition, it has also been determined that increasing concentrations of phosphate, silicate, and natural organic matter reduces the efficacy of

electrocoagulation [41]. On the other hand, bicarbonate, nitrate, sulphate, and chloride do not affect the elimination efficiency.

Electrocoagulation can be applied in situ and also ex situ. In ex situ technology, contaminated water passes between electrodes. The current causes arsenic to migrate toward the electrodes, and also modifies the pH and the redox potential of water causing arsenic to precipitate. Afterwards, solids are removed from the water by a filtration stage [40].

Arsenic electrocoagulation removal has been studied under various conditions of electricity intensity, pH, electrode connectivity, dissolved oxygen, temperature, and sedimentation. Several designs have been developed, but further research is necessary to better understand this process and make possible a significative increasing in arsenic removing [41], specially, the dependence of pH in function of the presence of other substances in the aqueous solution. The system pH affects the oxidation of Fe (II) and As (III). The presence of phosphate and silicate also has an effect by reducing the removal of arsenic, but it is not known how this process works.

### 2.1.4. Removal of As (III) and As (V) Using Iron-Rich Sludge

Amorphous or crystalline iron oxides have a high arsenic adsorption capacity. Removal of As (III) with a typical adsorbent is more difficult than the removal of As (V) due to the positive charge of the adsorbent surface and the neutral charge of As (III) in neutral conditions of pH. This is the reason why its extraction is pH dependent: the adsorption of arsenic decreases at high pH or in the presence of phosphates [42].

Most coal-mining drainage contains high proportions of iron and its treatment systems produce large quantities of iron-rich sludge whose disposal and reuse cause major environmental problems, so its use as an adsorbent has environmental and economic advantages.

The iron-rich sludge employed consists of goethite and calcite from the drainage treatment of coal mines; it provides an arsenic adsorption of almost 70%, after one hour of contact. After 8 h of contact, the adsorption increases up to 95%. However, after 48 h, the treatment improves just an extra 2–5%. The maximum adsorption capacity of As (III) and As (V) is in the range 67–72 and 21.5–24.6 mg/g adsorbent, respectively, at 293–323 K [37].

### 2.1.5. Fixed-Bed Sorption System Using Coconut Shells and An Iron Matrix

The removal of different potential toxic elements from gold-mine effluent can be carried out in fixed downflow bed columns using coconut shells and different iron solid matrix, such as iron-oxide-coated sand, ferric hydroxide-coated newspaper pulp, granular ferric hydroxide, or iron filings mixed with sand [38,40].

There are few studies in which agricultural and natural materials are used in continuous systems for the removal of potential toxic elements from effluents, so it is necessary to carry out pilot-scale research to demonstrate the applicability of biosorption processes in wastewater treatment on an industrial scale. Acheampong and Lens [38] showed the sorption system based on coconut shells' capacity for Cu (II) sorption. This method can be applied to remove As (V) in continuous and non-continuous bed conditions, by reusing in multiple stages of sorption and desorption. Figure 5 shows a schematic diagram of an arsenic adsorption pilot plan.

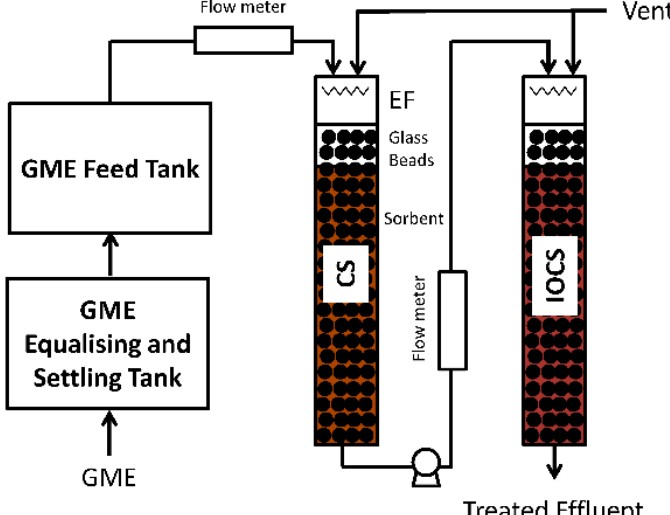

**Figure 5.** Schematic diagram of the two-stage pilot plant set-up (CS: coconut shell; IOCS: iron oxide coated sand; EF: effluent distributor; GME: gold mining effluent; FM: flow meter). Adapted from [38].

Coconut shell with a particle size between 0.5–1.4 mm is introduced in the first column, filling a bed height of 150 cm. First column manages to remove up to 100% of Cu and Fe, while the As remains in the effluent. In the second column, iron-oxide-coated sand with a particle size between 1.0–3.0 mm is introduced, filling a bed height of 150 cm. In both cases, the sorbent is placed between a layer of beads of glass on the top and a sieve placed on the bottom of the columns. The influent is pumped into the columns and passes through them. After the treatment, the concentrations of the different potential toxic elements obtained are measured. After 3 cycles in both columns, only As is still detected in the effluent. The optimum number of cycles is found to be 8. Other variables optimized are length of the bed, contact time, number of columns, sorption sites, and volume of gold mining effluent, and As is satisfactorily removed from contaminated water.

This installation demonstrates the ability of the coconut shell and iron-oxide-coated sand to remove arsenic and copper, as well as other heavy metals (Pb, Fe, and Zn) in order to comply with actual normative in the Netherlands. In addition, it demonstrates the capacity of the pilot plant for the treatment of large volumes of contaminated effluents.

### 2.1.6. Arsenic Removal with Biogenic Ferric Precipitates

Removal of arsenic can be carried out in situ by precipitating the iron present in the contaminated water. Iron oxides, such as jarosite, schwertmannite, or goethite can be applied in the removal of arsenic from mining effluents. A study of a biological treatment for acidic solutions containing iron [Fe (II)] and arsenic [As (III)] has been performed by Ahoranta et al. [39].

A diagram of the process is shown in Figure 6. It consists on a fluidized bed with biological iron oxidation reactor, followed by an iron precipitator for the subsequent separation of precipitates by gravity in a settler. Furthermore, experimental conditions have been optimized for arsenic removal. In this sense, the highest iron oxidation and precipitation rates were 1070 and 28 mg As/L for 5 and 7 h of retention time respectively, achieving a solid precipitate with 96–98% of Fe (II) present in affluent at pH 3.0. These conditions, after the treatment process, yielded to reach a 99.5% of arsenic removal efficiency.

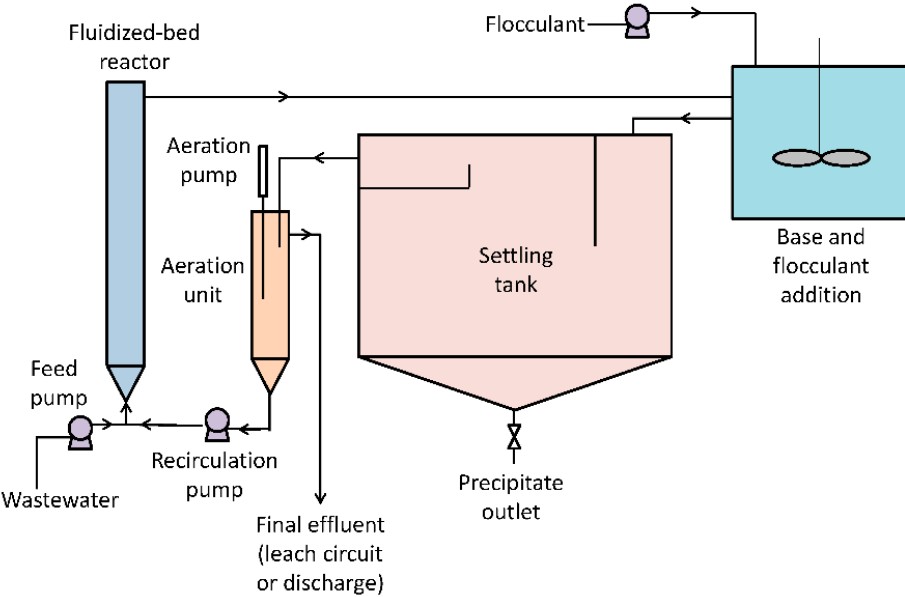

**Figure 6.** Schematic diagram of the experimental system. Adapted from [39].

### 2.1.7. Bioleaching of Arsenic Using *Acidithiobacillus thiooxidans*

In-situ or ex-situ bioleaching technology applied to contaminated sediments uses acidophilic and chemotropic bacteria to oxidize sulphur and Fe (II) under acidic conditions. This oxidation makes solubilization of potential toxic elements contained in the sulfides of the sediments possible.

Bioleaching using *Acidithiobacillus thiooxidans* is an effective method for the reduction of arsenic concentration from mine tailings with high concentrations of arsenic (approximately 34,000 mg/kg) [39].

The studies reported by Ahoranta et al. [39] were performed to determine the optimum conditions to bioleach the potential toxic elements present in batteries. The initial experimental conditions were: pH 1.8–2.2, temperature between 25 and 40 °C and solid concentration in the interval of 0.5–4.0%. Optimal conditions were set to pH value of 1.8, 30 °C, and 0.5% solids concentration. These conditions yielded to a maximum As removal efficiency of 47%.

The results showed that the leaching efficiency was similar regardless of pH. A higher leaching velocity was observed with low initial pH (1.8), which could be due to a greater initial fixation of the cells to the tailings. Furthermore, it was observed that changes in temperature and solids concentration affect leaching. Leaching efficiency decreased with an increase in temperature due to the lower bacterial growth. The highest initial leaching ratio was obtained at 25 °C. On the other hand, efficiency increased with a decrease in solids concentration. Researchers also observed that jarosite presented in tailings after leaching was higher when solids concentration was also higher due to the lower efficiency of arsenic bioleaching.

Experimental installation is usually based on a packed-bed column reactor where residual water with arsenic is pumped. When column saturates, arsenic is stripped and reactor is biologically regenerated. Some studies performed at The Center for Bioremediation at Weber State University have studied this process on a bench scale reactor. In this system, bacteria produce sulfuric, nitric, and organic acids that are important in process optimization. In addition, surfactants are produced because of the biological activity, enhancing arsenic leaching in the system [40].

### 2.2. Cadmium

Between all techniques studied, there is now a certain predominance towards the use of biotic techniques for its extraction, as opposed to other types of techniques such as soil washing, the use of activated and non-activated adsorbents, or absorption with zeolites.

### 2.2.1. Microorganism Mediated Biosynthesis of Cadmium Chalcogenides

This removal method consists in nanocrystals production of metal chalcogenides (CdS, CdSe, CdTe) based on the metabolic activity of living organisms [39]. Nanocrystal produced, such as Q-dots, are characterized to be semiconductors and, hence, they are suitable for developing cutting edge technologies including optical devices (optical storage, light-emitting diodes), solar energy conversion, or signaling of in vivo process as fluorescent labels [43]. In this process, bioremediation is based on the minimization of cadmium bioavailability because microorganisms do not decompose potential toxic elements.

Microorganisms cells surface contains chelates that allow cadmium to cross cell walls. Once inside the cell, cadmium is combined with $S^{2-}$, $Se^{2-}$ or $Te^{2-}$ to produce Q-dots chalcogenides. These particles can be separated from cellular solution by magnetic methods [43]. Biosynthesis is achieved in a modular reactor based on microorganisms trapped in mineral matrices that combine cadmium depletion and chalcogenides nanoparticles production.

The main disadvantage of this environmental-remediation technique is the difficulty with (i) Q-dots metal chalcogenides recovering, and (ii) biological manipulating. In this way, further research is needed in order to design bioreactors to minimize a possible environment contamination because of the low recovery of toxic Q-dots and to make biological manipulation safer.

### 2.2.2. Removal Using Dead Biomass

Bio-treatment for potential toxic elements removing has demonstrated to be an interesting environmental technology due to its low operating costs and high detoxification efficiency [44]. The use of dead biomass is proposed in order to overcome disadvantages detected in living biomass studies, such as the addition of nutrients and the maintenance of microorganisms, since they are sensitive to water quality (pH, oxidation-reduction state), but there are other problems of application in real scale, such as the difficulty of separating biomass from water after treatment and low mechanical resistance with long-term use. If the dead biomass is immobilized in a polymer matrix, its presence confers high resistance on chemical environments and provides other advantages such as efficient regeneration and ease of solution separation [45].

The effectiveness of immobilized dead *Bacillus drentensis* sp. use in a polymer matrix for the bio-sorption of metals in acid waters from underground mines has been studied [46]. Figure 7 shows the configuration of the pilot plant feasibility test. One of the most important issue to be considered is the immobilization matrix material, as it determines the mechanical strength, stiffness, and porosity characteristics [47]. Alginate, polyacrylamide, and polyvinyl alcohol has been used as bio-carrier packed layer but, recently, the use of polysulphone has provided good results for Pb (II) and Cu (II) removal, reporting maximum adsorption capacities of 0.3332 and 0.5598 mg/g, respectively. Hence, poysulphone material reduces the process costs [48].

Column tests were performed by adding different amounts of biomass to an artificial solution with different initial concentrations of several potential toxic elements (Cd, Cu, As, Zn, Pb, Fe) at pH 3.0. Polysulphone has a porous structure that adsorbs cadmium from the solid form at the boundary between the biomass and the matrix and/or the interior of the biomass. Elimination efficiency in lead and copper column tests is over 87%. The amount of lead absorbed per gram of bio-carrier is 1553 g, which shows the high sorption capacity of the elements used for the removal of potential toxic elements. In the pilot prototype, 80 tons of groundwater were contaminated with potential toxic elements for 40 days. The disposal capacity was, at least, 1098 tons per kilogram of bioculture.

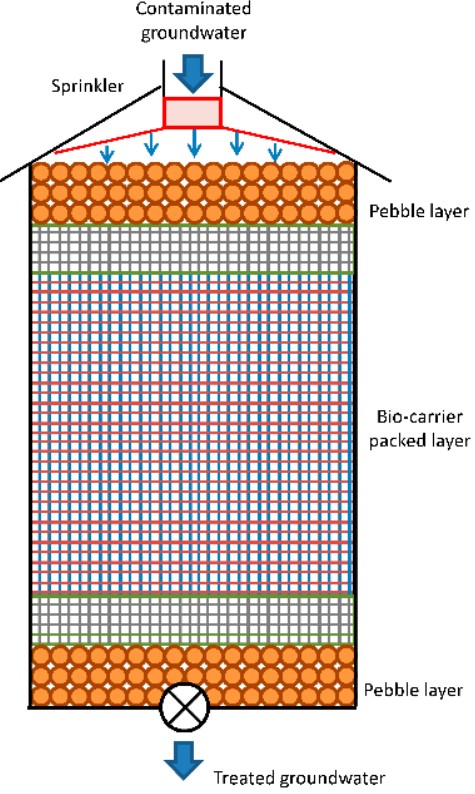

**Figure 7.** Configuration of the pilot plant feasibility test. Adapted from [39,46].

### 2.2.3. Removal by Nostoc Muscorum Cyanobacterium

There are many works on the removal of a single metal in aqueous solutions using cyanobacteria, but recent works such the one presented in [12] proposes the biosorption of potential toxic elements of a multicomponent system consisting of Cu (II), Zn (II), Pb (II) and Cd (II) present in an aqueous solution using *Nostoc muscorum*.

The choice of cyanobacteria is based on the fact that they are organisms that require a very low nutritional contribution and can survive under strict environmental conditions. The study was carried out using the statistically valid Plackett–Burman factorial design to better understand the biosorption of present metals. As main factors, the concentration levels of the four potential toxic elements (Cu (II), Zn (II), Pb (II) and Cd (II)) were chosen in the study.

The results show that Pb (II) is eliminated with high efficiency (96%), followed by Cu (II) (96%), Cd (II) (80%) and Zn (II) (71%). Lead shows an inhibitory effect on the removal of other metals, which is attributed to its small ionic radius in comparison with other metals. In addition, the values obtained from sorption show that the cyanobacteria *Nostoc muscorum* must be further studied because its potential in the disposal of contaminated waste water. Maximum elimination rates after 60 incubation hours for Pb (II), Cu (II), Cd (II) and Zn (II) are 96.3%, 96.42%, 80.04%, and 71.3% respectively. These results could be improved with a better process knowledge.

### 2.2.4. Adsorption with Activated and Non-Activated Carbons Prepared by Pyrolysis of Oily Sludges

The adsorption of potential toxic elements present in aqueous solutions with active carbon has been demonstrated to be an efficient and low-cost adsorption process [24]. The adsorbents, porous carbons obtained from oily solid pyrolysis wastes [49], were physically and chemically activated to improve their effectiveness in the removal of potential toxic elements from wastewater [50]. Carbon activation contributes to reduce the sludge volume and reduces the environmental costs and problems associated with the disposal of contaminated waste.

The adsorption of cadmium ions with active and non-active carbons has been achieved, reaching average adsorption rates between 78% and 98%. The most significant factors in cadmium removal were the initial metal concentration and the adsorbent dose used, 500 ml with 10 mg Cd/L and pH 6. Is has also been reported that oily sludge is a potential resource for the generation of carbonaceous adsorbents for the treatment of wastewater, but certain economic, technical and environmental aspects must be improved if it is to be used on a large scale.

### 2.2.5. Adsorption by Zeolites Synthesized from Rice Husks

Zeolites are effective for purifying water and gases [51–53]. There are numerous works on their production using coal ash [54–58]. However, there is hardly any research on the production of zeolites from the silica content of ash from rice husks. Investigations have been conducted on the use of synthesized zeolites from rice husk ash, the white part of which has a high content of silica that can be used as a precursor of zeolites for cadmium removal in aqueous solution.

The elimination capacity of Cd (II), was studied in [28] with a concentration between 50 and 500 mg/L. The maximum removal capacity of the cadmium ion of the Na-A zeolite is 736.38 mg/g and that of the Na-X zeolite is 684.46 mg/g. Maximum rates for zeolites Na-A and Na-X elimination are 98% and 97% respectively, for 9.2 g/L dosage and pH of 9.0. Na-A zeolite exhibits higher cadmium removal than Na-X zeolite because of the higher cation exchange capacity. These results were compared with those obtained with a commercial synthetic zeolite and an elimination capacity close to that of commercial zeolites was reported.

### 2.2.6. Sequential Soil Washing Technique

Soil washing is an ex-situ technique that is suitable for the removal of potential toxic elements from the ecosystem due to its high efficiency and profitability. A considerable amount of literature has been published on soil washing technique applied to potential toxic elements removal with a single reagent and/or in a single washing step [59–61]. In these studies, the fact of several metals coexisting in contaminated areas have pointed to the insufficiency of a single reagent and/or a single washing step as a successful contamination elimination.

A triplicate sequential soil wash test using a typical chelating agent (Na$_2$EDTA), an organic acid (oxalic acid) and a weak inorganic acid (phosphoric acid) has been carried out in [62] to evaluate how to clean soil contaminated with potential toxic elements (As and Cd) in the surroundings of a mining zone.

The results analysis shows that the primary components of arsenic and cadmium in the soil are arsenic residues (O-As) and exchangeable fraction, which represents 70% of cadmium and 60% of total arsenic. The different status as which potential toxic elements are present in soil makes a sequence of soil washing necessary. In Wei et al. [62], the optimum sequence of soil washing is phosphoric acid, oxalic acid, and Na$_2$EDTA with the following experimental conditions: Agitation speed, 150 rpm; liquid/solid ratio, 15/1; washing time, 30 min; room temperature; agent concentrations of 0.075 M for Na$_2$EDTA, 0.075 M for oxalic acid and 0.05 M for phosphoric acid, and the original pH of the washing agents. The results demonstrate that the tested six influencing factors except temperature had a marked effect on the efficiency of heavy metal removal.

The good elimination efficiency (41.9% for arsenic and 89.6% for cadmium) and the minimization of harmful effects of mobility and bioavailability of the present potential toxic elements were pointed to be under consideration for further study.

### *2.3. Mercury*

In the case of mercury, three conventional and two innovative technologies are reported here. As conventional methods, thermal desorption, chemical extraction, and solidification/vitrification are explained. As innovative methods, both a biological technique based on the use of an aquatic plant,

*Limnocharis flava*—present in wetlands, and a physical/chemical low-cost technique based on the use of a natural adsorbent—a nanobiocomposite hydrogel modified with glutaraldehyde, are reported.

### 2.3.1. Thermal Treatment Based on Mercury High Temperature Thermal Desorption

Thermal desorption is a no destructive technology based on the volatilization of water in land also with the mercury it contains. It is a non-destructive technology because neither water molecules nor metals or organics pollutants are destroyed during operation. In addition, mercury is not usually oxidized in this system. It is characterized to be an in-situ or ex-situ treatment. In ex-situ case, land must be excavated and, afterwards, it must be transported to the equipment installation location [63].

As it is explained by the United States Environmental Protection Agency, land is introduced into a heated desorber, where both a carrier gas or a vacuum system can be used to transport the volatilized mercury produced by heating to the gas treatment located downstream. This treatment consists in a simple particle removal system, such as a baghouse or a wet scrubber, followed by a mercury condensation step and a sulfur-impregnated activated carbon adsorption to adsorb the residual mercury. Finally, residual gas is emitted to the air. After being treated by thermal desorption, the soil is cooled by spray water and it is returned back to the site [64].

The mains design parameters to be controlled in this technology are the bed (desorber) temperature profile and residence time of the carrier gas.

The most common equipment used to heat the soil in desorber are screw units and rotary driers. Screw units transport the land along the process unit while a hot oil or steam stream circulates through small pipes located between the screw hollows. On the other hand, rotary driers are slightly inclined cylinders that can be directly or indirectly fired. In the directly fired option, fire is applied directly upon soil surface while the indirectly fired option is characterized by heating the heater surface by the use of flames located in a reactor placed in contact with the rotary drier. In both screw units and rotary driers, land is heated to 320–500 °C. The technology has proven it can produce a final contaminant concentration below 2 mg/kg, in some cases even lower than 0.05 mg/kg [63,64]. After soil sieving, land particles are introduced into an extractor. HCl (acid) is used as extractant agent when mercury is mainly present as Hg(0), while organic compounds, such as acetone, hexane, methanol, or amines are used when mercury is found to be bonded to other organic structures.

In in-situ operation mode, thermal desorption method is applied simultaneously by heating the soil with resistors (usually 400–600 °C) and using a vacuum for a gas emission. The soil may be heated with a steam high-flow at high temperatures through injection wells.

### 2.3.2. Chemical Extraction Mercury Removement

Chemical extraction consists in the use of an extractant agent (acid or organic compound) to separate mercury from soil because the high solubility of the metal in the extractant agent. Thus, the volume of the hazardous waste that must be treated is reduced. It is an ex situ and a non-destructive method.

Before chemical extraction, a physical soil separation into different particle sizes is often used: Thus, fine and coarse fractions are usually treated by separately. This step may result in the enhancing of extraction kinetics [65].

Residence time of soil in the extraction unit is generally on the range between 10 and 40 minutes, but it depends on the soil type and mercury concentration. The soil-extractant mixture is continuously pumped out of the extractor tank, and the soil and extractant are separated using hydro cyclones. Afterwards, soil is rinsed with water to remove entrained extractant agent and mercury. Finally, extraction solution and rinse water are regenerated by using NaOH and flocculants to remove mercury. Thus, the extractant agent is reformed and mercury is concentrated in a precipitate potentially suitable to be recovered.

Chemical extraction efficiency is usually over 90% and it could reach 99% if metals are not strongly bonded to other structures.

### 2.3.3. In situ Solidification/Vitrification Method

The objective of this method is to reduce the mercury mobility in the environment in both chemical and physical meanings by trapping it within its "host" medium. For the purpose of mercury remediating, an electric current is applied in order to melt soil at very high temperatures (1600–2000 °C). Consequently, once the soil is cooled down, mercury is immobilized in a vitrified glass and crystalline mass. After applying heat, water vapour, some mercury amount previously volatilized, and other compounds from organics pyrolysis are captured in the off-gas collection hood. Afterwards, off-gas is treated by removing particulates, the volatilized mercury, and other pollutants from the gas. Finally, vitrified soil is found to be chemically stable, leach-resistant, glass, and crystalline material similar to obsidian or basalt rock.

Soil composition, especially humidity, determines whether this technology is suitable to be applied or not. If silt and clay contents are high enough, water release between soil particles can be very difficult. In addition, a high soil porosity can make necessary a previous land compaction.. Additionally, humidity reduces the technology efficiency [65].

The main advantage of the solidification/vitrification method is the high mercury "removal", over 99%, but it has several disadvantages, such as (1) the land depth in which the treatment is suitable to be applied is small in comparison with other technologies (2) high cost, and (3) the land reutilization is limited as soil characteristics are completely modified [66].

### 2.3.4. Removal of Mercury from Gold Mine Effluents Using Limnocharis Flava In Constructed Wetlands

The mechanisms of potential toxic elements removal in horizontal artificial wetlands are complex and include binding to sediments and soils, precipitation and co-precipitation of metals as insoluble salts, plant absorption, and, to a lesser extent, microbial metabolism [67]. A schematic view of experimental pilot plant is shown in Figure 8.

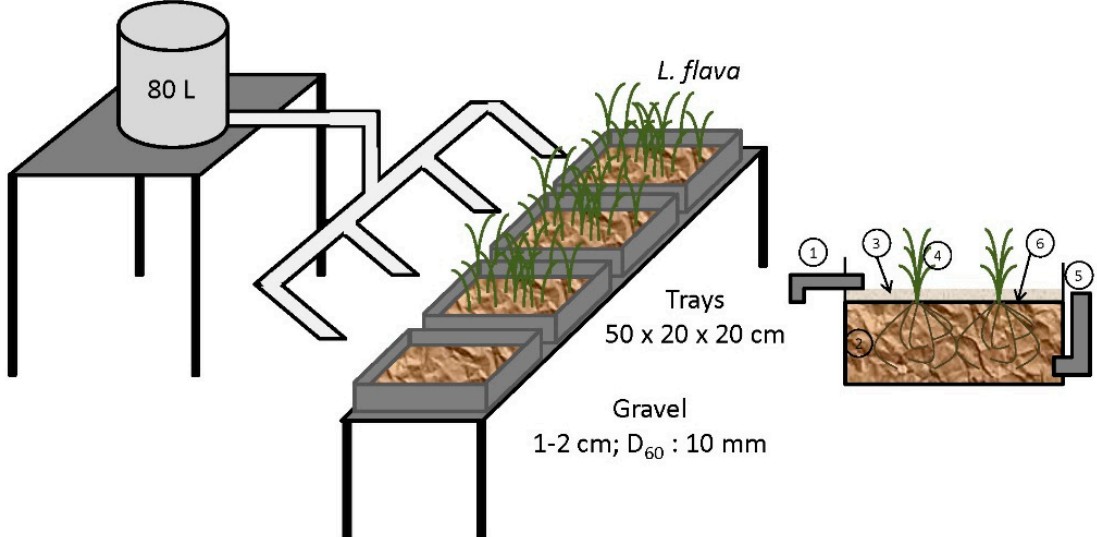

**Figure 8.** Schematic view of experimental setup in the pilot-scale constructed wetland system. Lateral section of planted systems *Limnocharis flava*: 1. Supply pipe; 2. Distribution area (coarse gravel); 3. Gravel 1-2 cm; D60: 10 mm; 4. *Limnocharis flava*; 5. Adjustable output pipe to control water level; 6. Water level. Adapted from [68].

The potential of the *Limnocharis flava* aquatic plant for the phytoremediation of water contaminated with mercury in a constructed wetland with horizontal underground flow was evaluated in [62] through a pilot experiment using an effluent from a gold mine in El Alacrán (Colombia) with a mercury content of 0.11 ± 0.03 mg/mL enriched with $HgNO_3$ (1.50 ± 0.09 mg/mL) [68].

The results obtained show a mercury concentration reduction only in the wetlands in which *Limnocharis flava* was used. The most important variable for mercury removing is the exposure time. Mercury concentration was reduced from 1.50 ± 0.13 to 0.15 ± 0.04 mg/mL after 30 days. With exposure time optimization, removal yield increases up to nine times relative to the control experiment.

The *Limnocharis flava* macrophyte has a high potential for the phytoremediation of water contaminated with mercury, reducing its concentration in contaminated water by 90%. The construction of this type of wetland is a viable option for the treatment of effluents from gold mines rich in mercury. Construction of this system is planned in El Alacrán (Colombia) in order to optimize it and evaluate its performance for several years before its complete construction.

### 2.3.5. Removal of Hg (II) Ions from An Aqueous Environment Using Glutaraldehyde Crosslinked Nanobiocomposite Hydrogel Modified by TETA and β-cyclodextrin

A study of Hg (II) ions removal from aqueous solutions using a new nanobiocomposite hydrogel modified by triethylene tetramine (TETA) and β-cyclodextrin has been reported to be an alternative and cost-effective technique for the remediation of mining wastewater [69].

The nanobiocomposite hydrogel is composed of chitosan, vegetable gum, montmorillonite, and zinc oxide nanoparticles crosslinked by glutaraldehyde and modified by TETA and β-cyclodextrin for an effective removal of Hg (II) ions from wastewater. This reduction is based on the availability of amine and hydroxyl residues for enhancing their interaction with glutaraldehyde.

The maximum elimination values of Hg (II) achieved by TETA and β-cyclodextrin are 97.9% and 70.1%, respectively. In absorption case, results reported by TETA and β-cyclodextrin are 407.9 mg/g and 292.1 mg/g, respectively. The main experimental used parameters are: pH, 6.0; contact time, 6 h; initial mercury concentration, 250 mg/L; and biomass dosage, 0.6 g/L.

Results analysis reported a higher adsorption of Hg (II) yield using TETA-cyclodextrin because of its greater surface roughness, lower compaction, and greater availability of functional groups. In addition, maximum mercury removal from mine wastewater was set at 80%. This result was obtained with a maximum column height of 12 cm, a minimum flow rate of 1 ml/min, and maximum dilution of 50%.

## 3. Discussion on the Systematic Review

All the techniques studied present adequate results for their application at a pilot scale. Some of them have pilot-scale projects that support the results obtained in the laboratory. The studies at pilot scale are necessary in order to identify the adjustments for their real-scale implementation. Table 4 shows a summary of the main studies considered in this work.

**Table 4.** Summary table of the reviewed studies.

| Metal/Metalloid | Treatment | Articles | References |
|---|---|---|---|
| **Arsenic** | Adsorption | 2 | [34,35] |
| | Electrocoagulation | 1 | [41] |
| | Precipitation | 1 | [36] |
| | Bioleaching | 5 | [38–42] |
| **Cadmium** | Adsorption | 11 | [24,28,50–58] |
| | Sequential soil washing | 4 | [59–62] |
| | Biosynthesis of metal chalcogenides | 7 | [12,39,43–47] |
| **Mercury** | Phytoremediation | 2 | [62,67] |
| | Nanobiocomposite hydrogel | 1 | [69] |
| | Others | 4 | [63–66] |

Between the techniques here explained, very efficient methodologies for arsenic, cadmium, and mercury removal can be found. However, in each case, not only a specific determination as to the applicability of the principles and provisions of each technology must be made, but also economic

issue is an important factor to determine final decision. To make a closer discussion about this issue, Table 5 reports a costs summary of different remediation technologies for metals removing based on the USEPA studies [63]. The methodology followed in USEPA studies allows to classify the technologies into two scenarios: large and small sites. Hence, the scale up effect can be observed.

**Table 5.** Costs summary ($€/m^3$) of different remediation technologies. Source: [63].

| Technology | Large Site Cost | Small Site Cost | Average Cost | In-Situ/Ex-Situ |
|---|---|---|---|---|
| Chemical extraction | 320 | 1468 | 894 | "ex situ" |
| Phytoremediation | 214 | 1312 | 763 | "ex situ" |
| Adsorption | 303 | 405 | 354 | "ex situ" |
| Bioleaching | 105 | 145 | 125 | "ex situ" |
| Solidification | 58 | 178 | 118 | "in situ" |
| Soil washing | 62 | 166 | 114 | "ex situ" |
| Thermal desorption | 63 | 148 | 105 | "ex situ" |
| Electrocoagulation | 83 | 104 | 93 | "in situ" |
| Precipitation | 76 | 91 | 84 | "ex situ" |
| Bioremediation | 27 | 89 | 58 | "in situ" |

From inspection of Table 5, as it was expected, ex situ treatments present, on average, higher associate costs than in situ ones. One of the main reasons of this finding is associate to the costs implied in land excavation and soil transportation. For the sake of comparison, technologies can be classified by treatments groups as thermal, physical/chemical, and biological treatments. A closer observation of Table 4 shows that bioremediation emerges as the best interesting methodology to metals remove. By contrast, phytoremediation is observed to be one of the most expensive methods. Most of physical/chemical treatments costs are situated in a compromise cost place (e.g., soil washing, adsorption, electrocoagulation, precipitation) but chemical extraction is catalogued to be the most expensive treatment because of the need of a specific treatment for the extracting agent to be used afterwards. Finally, thermal methodologies (e.g., solidification, thermal desorption) can be interesting from an economic point of view, when application is possible, because the in situ application.

### 3.1. Arsenic

In arsenic removal, it is important to determine the arsenic's speciation, since As (III) is more toxic, more mobile, and more difficult to eliminate by using many of the treatment technologies previously explained than As (V) is.

In the systematic review carried out in the previous section, two predominant lines have been evaluated; the most common techniques are those based on the use of iron compounds. However, those that eliminate arsenic through the use of bacteria have also been found interesting for mining contaminated waters.

There are few studies that compare co-precipitation and adsorption by iron minerals. Also, few studies evaluate the design of arsenic's mechanism of natural attenuation.

If adsorption-co-precipitation rates are observed, the rate of arsenic adsorption is found to be 0.90 mM higher than arsenic-ferrihydrite co-precipitation. No significant differences are found with schwertmannite use. The adsorption with goethite reports better results than co-precipitation. The reason is currently still unknown, and so more research is required. Arsenic adsorption by schwertmannite seems to be more effective than for the other iron minerals [34]. The As adsorption by schwertmannite included exchange reactions between sulfate and As (V) [70].

The elimination of As (III) by electrocoagulation with iron electrodes depends on Fe (II) oxidation and Fe (III) precipitation, so it also depends on the operating parameters affecting Fe (II) oxidation, such as current intensity, pH, initial concentration of arsenic, or the presence of other ions in the aqueous solution [41].

Another disadvantage of arsenic removal by electrocoagulation is the only partial oxidation of Fe (II) to Fe (III). In addition, the presence of phosphates and silicates adversely affects arsenic removal in Fe (II) and Fe (III) co-precipitation, similar to electrocoagulation systems with iron electrodes [41]. However, one advantage of electrocoagulation with iron electrodes is that As (III) conversion to As (V) can be achieved without the use of any chemical reagent.

Arsenic [III and V] removal by using iron rich sludge can be achieved by a sludge composition of goethite and calcite, according to X-ray diffraction (XRD) analysis. It also contains other iron oxides [37]. The possibility of reuse the adsorbent in several consecutive cycles has been also studied. The adsorbent adsorption-desorption capacity decreased by about 10% in each cycle. As a result, iron-rich sludge technology can be used effectively at an average of three times [37].

Adsorption systems based on agricultural and natural fixed-beds have been successfully used for zinc and nickel removal. In this case, a low effectivity was detected in water treatment from electroplating using sugarcane bagasse, probably due to electrostatic repulsion forces between the cationic surface of the bagasse and the metal ions [71]. However, when coconut shell was used, this situation was not detected because it was negatively surface charged at the work pH of 6.5. As a consequence, potential toxic elements with positive charge were removed successfully because of attractive forces between the sorbent and the sorbate [38].

Removal of arsenic present in acid solutions by biogenic ferric precipitate has also been carried out, but the stability or storage of arsenic-containing precipitates was not studied [38]. The main inconvenience in using this technique is the discrepancy observed between the results for the arsenic present in the liquid phase and for that present in the solid phase due to an overestimation of the total concentration of arsenic in the effluent [39]. However, this inconvenience could be avoided by the use of a graphite furnace or inductively coupled plasma emission spectrometry (ICP-ES) since the detection limits are lower [39].

Finally, bioleaching of arsenic using *Acidithiobacillus thiooxidans* make solubilization of potential toxic elements contained in the sulfides of the sediments possible. In addition, solids concentration was observed to be inversely proportional to the elimination efficiency and directly proportional to the time at which the leaching begins [72]. However, it was found some corrosion problems of mine tailings produced by bacterium [72].

## 3.2. Cadmium

Removal of cadmium from a contaminated matrix can be carried out through a variety of biotic or abiotic techniques. These techniques must be studied in each particular situation because their high sensitivity to environment conditions.

Conventional techniques for the treatment of cadmium contaminated water are not characterized to be economically profitable for small and medium sized industries (see, for example, chemical extraction for small sites in Table 4). This is the reason why innovative techniques with natural materials are being studied for decontamination.

A very little amount of literature has been published on the metal chalcogenides biosynthesis using encapsulated microorganisms for cadmium removal. However, findings reported by Vena et al. [43] suggest this technique to be interesting in this field, for its ability to reuse waste as functional nanomaterials, so further research must be carried on.

The results obtained from the extraction of potential toxic elements with *Bacillus drentensis* sp. indicate the convenience of using polysulphones as a bio-carrier packed layer adsorbent because its high porosity and low cost. As in the metal chalcogenides case, there is little amount of literature published on the item of cadmium removal with *Bacillus drentensis* sp. [46]. The findings of this study suggest that cadmium removal by *Bacillus drentensis* sp. could be a good solution for water treatments. In this sense, further research in the field is necessary.

The kinetics of elimination of potential toxic elements with the bacterium *Nostoc muscorum* conforms to the second-order kinetics described in the existing literature. [12] and given as follows:

$$\frac{1}{Q_t} = \frac{1}{k_2 Q_e^2} + \frac{1}{Q_e} t \tag{1}$$

where $k_2$ ($g^{-1}$ $mg^{-1}$ $min^{-1}$) is the rate constant of the pseudo second-order sorption and $Q_e$ and $Q_t$ denote the amounts of metal ions sorbed at equilibrium and at time $t$ (mg $g^{-1}$), respectively. The metal removal system in a multicomponent system is complex and unique for each combination of metal concentration and microbial strain employed in the process [12]. For this reason, the study of one of these metals and microorganisms cannot predict its elimination in a multicomponent system [73].

Removal of cadmium present in aqueous solutions with carbonaceous adsorbents are more efficient when activated carbons (96%) are used than in the case of non-activated carbon adsorbents used (20%) [50]. The values predicted by the Taguchi strongly differ from those obtained in experiments performed for both adsorbents, with an error of 11% for non-activated carbon and 7% for activated carbon. Therefore, the confirmatory experiments were carried out under the optimum expected conditions and the percentage of cadmium adsorption obtained in the confirmatory experiments was adjusted to the expected amounts. The preparation of porous carbons from oily solid waste contributes to reduction of sludge volume, costs, and environmental problems associated with its elimination. Adsorption cadmium efficiencies up to 98.28% can be obtained with active carbons use.

Cadmium adsorption by zeolites synthesized from rice husk eliminates Cd (II) by varying its concentration between 50 and 500 mg/L, while temperature (30 °C), solution pH (7.0) and cadmium dose (2.0 g/L) are keeping constant. Results show a cadmium removal efficiency of 97–98% after the previous treatment. Both Na-A and Na-X zeolite present similar results of that obtained with synthetic zeolites, such as 3A [38]. However, adsorption capacity of the Na-A and Na-X zeolites is much higher than that of other low-cost adsorbents such as fly ash [74], Egyptian kaolin clay [75], or petroleum shale ash [76]. As a result, cadmium removal by zeolites synthesized from rice rusk are presented to be a promising technique for cadmium sustainable reducing.

Finally, the results obtained for removing cadmium by soil sequential washing suggest that cadmium bioavailability in soil must be optimized to enhance cadmium removal. Some studies point that metals increasing solubility due to washing can be attributed to the dissolution of the remaining fraction and the transformation of other metal fractions. However, this result differs from previous studies in which amorphous Fe/Al oxide dissolution increases the bioavailability of cadmium in soils [77]. Cadmium removal efficiency near 90% suggests soil washing to be a promising technique to be optimized with further investigation.

### 3.3. Mercury

Five technologies for mercury removing were commented on in previous sections: Three could be classified as *conventional technologies* (thermal desorption, chemical extraction, and solidification/ vitrification) as their used has been more broadly tested. In contrast, two *innovative techniques* were also commented on: The use of the *Limnocharis flava* macrophyte in constructed wetlands for the treatment of gold mine effluents, and the use of a new hydrogel nanobiocomposite.

Thermal desorption is considered to be a good method for soils remediation as the final metal concentration can reach a value even lower than 0.05 mg/kg. However, thermal techniques are not usually the most economic ones between those possible to be used. As it is shown in Table 4, average thermal desorption costs can be established at 105 €/m³ (landfilling of activated carbon particles included).

Chemical extraction is characterized by a high efficiency of Hg removal (over 90%) but this efficiency is very sensitive on the different Hg combinations with other compounds present in soil. In addition, from Table 4 it can be concluded that chemical extraction is a very expensive technology.

In this sense, using chemical extraction is only reserved for situations where other technology cannot possibly be applied.

Solidification/vitrification presents a very high efficiency for Hg removing, as this is usually over 99% of initial Hg concentration, but the main disadvantage it presents is the change of the soil characteristics that can hinder some further uses.

Constructed wetlands are considered a viable alternative because of their low maintenance and operation costs. In addition, mercury concentrations are reduced by 90% if *Limnocharis flava* is used. These results are similar to those obtained by other authors who used species such as *Typha domingensis* [78], *Myriophylhum aquaticum*, *Ludwigna palustris*, and *Mentha* aquatic, but other authors report smaller efficiencies: 49% [79], 78,2% [80], and between 25% and 50% [81].

Mercury absorption by aquatic plants is affected by various factors, such as pH, temperature, mass flow, solar radiation, presence of chlorides, sulphates and phosphates, dissolved oxygen, biological oxygen demand, total organic carbon, total dissolved solids, and total suspended solids [68]. In the analyzed article [68], only the concentration of mercury at which *Limnocharis flava* is exposed and the exposure time have been studied in detail. The transfer coefficient (TC) for background mercury concentration ($0.04 \pm 0.01$ μg/g) was 0.02 mL/g, and a maximum value of 53.44 mL/g was reached after an exposure of 30 days, corresponding to a concentration of Hg of $8.02 \pm 1.14$ μg/g in the aerial part of the plant. On the other hand, using *Typha domingensis*, a TC of $7,751 \pm 570$ mL/g and an initial concentration of Hg of $9.0 \pm 0.4$ mg Hg/L were obtained after 27 days [78].

The effectiveness of the removal of Hg (II) ions from the aqueous medium using a nanobiocomposite hydrogel crosslinked with glutaraldehyde modified by triethylene tetramine (TETA) and β-cyclodextrin through the realization of six cycles of metal removal has been demonstrated. The effectiveness decreases as the cycles are repeated because the surface of the biosorbent becomes damaged. It is necessary to carry out experiments on the biosorbents reuse and regeneration, so that they can be used on an industrial scale in a sustainable way [69]. Finally, although bioremediation presents a lower efficiency than conventional techniques, the reduction in process costs (e.g., 118 €/m$^3$ for solidification vs. 58 €/m$^3$ for bioremediation) makes it more attractive for implementation and further investigation. The new nanobiocomposite hydrogel modified by TETA and β-cyclodextrin has proved to be an ideal and profitable agent for the remediation of contaminated water. When the hydrogel is modified with TETA-cyclodextrin, it has better elimination efficiencies than when modified with β-cyclodextrin. The optimization of these removal characteristics must be further investigated because of its good results.

## 4. Future Prospects

In order to guarantee the sustainability of human life on Earth, research on applications of waste treatments are mandatory. In particular, the EU is especially concerned about the importance of the transition towards a circular economy model and, thus, it has been developing a legislative framework for promoting it. Not only environmental consequences are involved, but a significant impact on the EU members' economies must also be considered. Such a transition presents an opportunity to transform the economy and generate new and sustainable competitive advantages for Europe [82].

In a number of end-uses, waste minimization involves using renewable material and energy [83]. Moreover, as [84] points out, by promoting the use of energy efficient equipment and renewable energy technologies, and also adopting measures for reduction of carbon footprint, the concern for hazardous wastes is also addressed in direction of long-term sustainability. Following the aforementioned, this paper intends to bridge the gap between the main techniques for arsenic, cadmium, and mercury disposal from hazardous wastes and locally available energy sources and sustainability.

In this sense, novel techniques involving biological treatments seem promising. Apart from the bioleaching or biosynthesis described in the previous section, microbial fuel cells (MFCs) and microbial electrolysis cells (MECs) can also be applied as effective treatments or as a complementary treatment with success.

A MEC is a device capable of converting the chemical energy contained in wastewater into hydrogen while reducing its organic load with an input of electricity [85]. In a MEC, electrochemically active bacteria oxidize organic matter and generate $CO_2$, electrons and protons in such a way the bacteria transfer the electrons to the anode and the protons are released to the solution. The electrons then travel through a wire to a cathode and combine with the free protons in solution thanks to an externally supplied voltage (higher than 0.2 V) and biologically assisted conditions [86].

On the other hand, MFCs can be defined as devices able to transform chemical to electrical energy via electrochemical reactions involving biochemical pathways [87]. To do that, these devices use electroactive microbes to degrade organics and produce electricity. These microbes are more sustainable and more durable compared to selective enzymes used in EFCs (enzymatic fuel cells), although they have worse electrochemical catalytic performance [87]. To produce electricity, a bacterial metabolic activity (the microorganisms donate electrons to an anode while oxidizing organic/inorganic waste) is produced in the anodic compartment while, separated by a membrane, electron acceptor conditions are shown in the cathode. In the oxidation reaction $HCO_3^-$, $H^+$, or $CH_4$ can be produced as co-products [88,89].

MECs and MFCs have shown a great potential for complementing costly wastewater treatment systems, being self-sustainable or even having a net positive energy output while pollutants are removed. Thus, interest is MECs and MFCs has been growing exponentially since the beginning of the 21st century although very few industrial applications exist nowadays due to the complexity of the maintenance of the proper conditions for the microbial population and the very low current rates. Although this technology is still not at an industrial-scale phase, some authors [90–93] point out its capacity to remove a mixed concentration of potential toxic elements from a contaminated influent, as the metal electroplate in the cathode chamber, with the advantage that they have very low energy needs. Moreover, as cells operate as MFCs, they can have a positive energy balance or, in the case of MECs, they can produce hydrogen or methane that can be used as bio-fuels [85,94]. Nevertheless, some large-scale applications are shown in [95,96].

These systems can be applied not only to remove organic waste from waste waters, but they have also shown good performance for electrochemical struvite precipitation [97], recovery of cobalt [90,91], or other heavy metals such as Cu or Ni [92].

In [96] a costs and revenues analysis both for MECs and MFCs is provided. The authors state that for a MEC, the revenue per kg of COD (chemical oxygen demand) removed can be considered constant if constantly applied voltage and constant cathodic Coulombic efficiency remained, and the costs decreased with decreasing internal resistance. For MFCs, however, the costs decrease with increasing current density. However, the revenues decrease with increasing current density for constant internal resistance. Thus, the highest revenues are obtained at the lowest internal resistance as the cell voltage is higher compared to higher internal resistances. In general terms, authors in [96] estimate the revenues of both systems in 0.35 €/kg (of COD removed) for the treatment of wastewater.

## 5. Conclusions

In this work, the application of the main hazardous waste management techniques in mining operations and in dumping sites has been reviewed. The systematic review has focused on the elimination of As, Cd, and Hg which are of particular interest of several European countries' directives, especially in Spain.

The development of biological methods is fundamentally due to the fact that conventional methods have higher costs, generate sludge, and may be inefficient. The use of biomass is increasingly being studied because it is an ecological, efficient, and technically feasible technique.

There is still hard work to do in the management of hazardous waste. What the treatments described have in common is that they are expensive, need long treatment times, and demand high rates of energy and/or chemical products. Moreover, they entail lots of human, machine, and energy resources in extraction and transport and in the restoration of the contaminated land fields.

In particular, future research lines must be focused not only on the improvement of the treatments' removal performance, but also on their energy requirements, as this can be one of the most limiting factors in carrying out treatment in a feasible way. Thus, apart from the improvement of current treatments, engineers and scientists must aim to remove potential toxic elements and other poisonous substances directly from leaches, as they constitute the main contamination vector. This objective is particularly relevant to the case of dumping sites, where leaches are extracted with efficacy thanks to drainage systems, but sophisticated post-treatments, energy-costly drying processes, and special storage management are needed.

The prospective analysis points out that biological treatments seem to be promising because of their effectiveness and their low energy consumption. In this item, the use of microorganisms for effluents decontamination is a promising and reversible technique that needs improvements—for example, in reactors design—for its further implantation. Moreover, the integration of MFCs or MECs as treatments to remove potential toxic elements may add other advantages such as the production of bio-fuels or positive energy balances.

**Author Contributions:** This paper is the result of the joint work of the authors. Conceptualization, A.G.-M. and A.B.-S.; Review, R.T.-F. and M.d.S.-M.; Analysis, R.L. and M.d.S.-M; Investigation, R.T.-F. and A.G.-M.; Methodology, A.B.-S., R.L. and M.d.S.-M.; Supervision, A.B.-S. and A.G.-M.; Validation, A.B.-S. and R.L.; Writing—original draft, A.G.-M. and M.d.S.-M.; Writing—review & editing, A.G.-M.

**Funding:** The APC was funded by Laboratorio de Inspección Técnica de la Escuela de Minas (LITEM), Universidad de León (Spain) and MDPI.

**Acknowledgments:** The authors want to thank all contributors to the project, and to the editors and reviewers for their valuable comments to increase the overall quality of the manuscript.

**Conflicts of Interest:** The authors declare no conflict of interest. The founding sponsors had no role in the design of the study; in the collection, analyses, or interpretation of data; in the writing of the manuscript, and in the decision to publish the results.

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
