# Peer review of "Remediation of Potential Toxic Elements from Wastes and Soils: Analysis and Energy Prospects"

_sustainability, doi:10.3390/su11123307_

Reviewer 1 Report

The authors present the review focused on removal of As, Cd and Hg from contaminated soil or wastes. The review deals only with removal techniques when the elements are removed from the matrices and not only immobilized. The review has good quality and can provide valuable information for potential readers. I can only recommend the authors to consider the use of tem “heavy metals” because As in not a metal.  The term toxic elements seems to be more suitable.

Author Response

REVIEWER 1 COMMENTS

The authors present the review focused on removal of As, Cd and Hg from contaminated soil or wastes. The review deals only with removal techniques when the elements are removed from the matrices and not only immobilized. The review has good quality and can provide valuable information for potential readers. I can only recommend the authors to consider the use of term “heavy metals” because As in not a metal.  The term toxic elements seem to be more suitable.

We thank the reviewer for his/her positive appreciation of the proposed manuscript and his/her valuable comments. Changes between the original submission and the revised version are highlighted in the ManuscriptTrackChanges document.

Regarding the use of the term “heavy metals”, it has been replaced by the proposed “potential toxic elements” in the whole paper.

The authors would like to express their sincere gratitude to the reviewer for his/her time and suggestions, as well as for his/her commitment to improve the manuscript. His/her advice and recommendations throughout the review round have been very beneficial and valuable.

Best regards,

THE AUTHORS.

Reviewer 2 Report

The manuscript “Remediation of hazardous wastes and contaminated soils with arsenic, cadmium or mercury: A systematic review and analysis of energy prospects” provides a number of information that may be interesting for the scientific community. The manuscript should undergo moderate English editing, please address this during revision. In this respect, I have appended below a list of major and minor concerns, the authors could address before publication. Therefore, without this clarification, it is difficult for me to recommend the manuscript for publication in its present form in sustainability.

Generally comment: Potential toxic elements or metals instead heavy metals, as indicated in a previous editorial of Sustainability (i.e. https://doi.org/10.3390/su10030636)”

 Line 34 - Missing quote.

 Line 52 - Which countries? Why did they increase the waste production?

Line 58 - Where?

 Line 73 - Is that your definition? In each case the quote is missing.

 Line 79 - Please quote the definition of leaching and percolate.

 Line 95 - pH?

 Line 99 - You referred to metals or in general? Please, you have to specify that.

 Line 113 - In literature there are a lot of paper about in-situ treatments.

 Line 117 - Bioremediation pathways could be more toxic than untreated ones.

 Line 132 - the authors have already written.

 Line 145 - “Observed by…”? Quote.

 Line 169 - “µg of… ??”

 Line 174 - 0.16 ppm? In water or soil? It’s not clear.

 Line 189 - the authors have already written that mercury is toxic in line 193.

Line 195 - the authors should write a definition of “bioavailability”.

 Line 203 - Who recommend 1 µg of Hg (II)/L in drinking water?

 In the first paragraph there are few quote for a review.

Figure 3 - You wrote that the period of publication was 2004-2017. Why is 2018 in figure?

 Table 3 - Please, correct Bioleaching from “-e” to “- “.

 Line 283 - … Who did determine the reduction of the treatment efficacy?

 Line 287 - … And about in situ technology?

 Line 294 - In line you wrote that the optimum pH for As (III) is 7. You have to specify that this pH is optimal in absence of other substances.

 Line 310 - Please use an approximately form, for example 67 mg/g

 Line 311 - mg of As/g adsorbent?

 Line 320 – Acheampong and Lens […]

 Line 336 – What kind of variables? What was optimal number of cycles for As removal?

 Line 339 – Actual normative… in which country?

 Line 351 – Please correct pH value to 3.0; 99.5% after?

 Line 368 – As removal.

 Line 384 – “Removing cadmium from water is fundamental for human health”. As and Hg removals are not important for human healt? Please delete this sentence.

 Line 401 – the authors wrote in line 394 that this bioremediation approach decreases the bioavailability (of cadmium?). But if cadmium is less bioavailable, why toxicity will increase? Please, specify.

 Line 409 – the authors mean “problem of application in real scale”?

 Line 415 – Please remove “by”.

 Line 444 – How much high is the efficiency?

 Line 459 – as line 310.

 Line 460 – Please, specify the dose used.

 Line 465 – Quote these works.

 Line 490 – Why was this soil washing sequence better than others? 

 Line 499 – Why did you investigate about only 2 treatments for Mercury? For others you cited more works. You have to insert at least another 2 treatments (conventional techniques for example).

 Line 522 – How much is the time of exposure?

 Line 538 – Elimination of?

 Line 542 – Adsorption of?

 In the second paragraph, you should insert a table with the discussed works and related results.

 Line 551 – Efficient than? You done an economical analysis?

Line 552 – I disagree because for example soil washing increase the remediation waste.

Line 565 – How much higher?

Line 568 – Is there a reason?

Line 606 – How much cost conventional treatments? As line 551. If you don’t, please report a quote.

Line 611 – Why could this technique be interesting?

Line 615:619 – You are discussing about Cadmium, please remove discussion about Pb and Cu.

Line 623 – Please quote the literature. You should insert the equation of kinetic.

 Line 629 – What are the predicted values?

 Line 638 - … removal after?

 Line 645 – Bioavailability of soils?? Maybe, you want to write “… cadmium bioavailability in soil must be optimized to enhance cadmium removal”.

 Line 648 – As line 645, it’s incorrect! The bioavailability is the accessible amount of pollutant (in this case cadmium) to organism.

 Line 659 – How much smaller?

 Line 662 – Remove BOD abbreviation because it’s useless for 1 time.

 Line 667 – Please correct 7751 to 7,751.

 Line 668 – 9.0 g of Hg(III)/mL of? Water?

 In the third paragraph, you have to compare conventional Hg treatments with innovative techniques.

 Line 688:690 – Can the authors include a cost-benefit analysis?

 Line 707:715 – It’s too vague. The authors have to explain better the MFC and MEC treatments, especially the way to produce hydrogen and methane. If it’s possible, you can also include an estimate of the revenue.

 In the fourth paragraph the authors can’t write about a “real” energy analysis because as you wrote in line 692 “… future research lines must be focused … on their energy requirements”. If you can’t do as I wrote for line 688, energy analysis could be integrated in conclusion at line 723 rather than repeat the result discussed in third paragraph.

Author Response

REVIEWER 2 COMMENTS

The manuscript “Remediation of hazardous wastes and contaminated soils with arsenic, cadmium or mercury: A systematic review and analysis of energy prospects” provides a number of information that may be interesting for the scientific community. The manuscript should undergo moderate English editing, please address this during revision. In this respect, I have appended below a list of major and minor concerns, the authors could address before publication. Therefore, without this clarification, it is difficult for me to recommend the manuscript for publication in its present form in sustainability.

General comment: Potential toxic elements or metals instead heavy metals, as indicated in a previous editorial of Sustainability (i.e. https://doi.org/10.3390/su10030636)

We thank the reviewer for his/her positive appreciation of the proposed manuscript and his/her valuable comments. At this point, according to the reviewer’s remark, the manuscript has been thoroughly revised by an English native-speaker colleague both in grammar and style. Figures and Tables have been also revised and significantly improved and, to our concern, the readability of the manuscript has significantly improved. Changes between the original submission and the revised version are highlighted in the ManuscriptTrackChanges document. We really want to thank the reviewer for his/her valuable comment which has contributed to increase the overall quality of the manuscript and we expect it now fulfills his/her quality standards.

Regarding the use of the term “heavy metals”, it has been replaced by the proposed “potential toxic elements” in the whole paper.

Line 34 - Missing quote.

We kindly appreciate the reviewer comment and agree with it. Thus, the proper quote has been added.

Line 52 - Which countries? Why did they increase the waste production?

We kindly appreciate the reviewer comment and agree with it. Thus, the requested information has been provided.

Line 58 - Where?

We kindly appreciate the reviewer comment and apologize for the inconveniences. In this case, it has been specified that the country is Spain.

Line 73 - Is that your definition? In each case the quote is missing.

We kindly appreciate the reviewer comment, but according to the recommendations given by reviewer 3, the full paragraph has been removed.

Line 79 - Please quote the definition of leaching and percolate.

We kindly appreciate the reviewer comment and agree with it. The definition of percolation has been incorporated to manuscript and a quote has also been provided.

Line 95 - pH?

We kindly appreciate the reviewer comment and agree with it. It is explained that metals coexist in acid water solutions.

Line 99 - You referred to metals or in general? Please, you have to specify that.

We kindly appreciate the reviewer comment and agree with it. It has been clarified in text.

Line 113 - In literature there are a lot of paper about in-situ treatments.

We kindly appreciate the reviewer comment and agree with it. The sentence has been modified in order to avoid confusions in this item.

Line 117 - Bioremediation pathways could be more toxic than untreated ones.

We kindly appreciate the reviewer comment and agree with it. The sentence has been clarified and paragraph has been improved.

Line 132 - the authors have already written.

We kindly appreciate the reviewer comment and agree with it. The sentence has been removed and paragraph has been improved.

Line 145 - “Observed by…”? Quote.

We kindly appreciate the reviewer comment. It is not a referenced observation but an author's conclusion. Thus, the paragraph has been rewritten in the following terms:

"The authors observe that a high energy consumption of certain treatments can constitute a real barrier for their effective application as can make them unaffordable due to the associated costs for the energy supply and the impossibility of supplying the required energy just by using renewable energy sources, such as photovoltaic systems, with an average specific energy capacity of about 170 W/m2."

Line 169 - “µg of… ??”

We kindly appreciate the reviewer comment and agree with it. It has replaced 10 μg/l por 10 μg As/l.

Line 174 - 0.16 ppm? In water or soil? It’s not clear.

We kindly appreciate the reviewer comment. It has been clarified that the concentration is in the soil.

Line 189 - the authors have already written that mercury is toxic in line 193.

We kindly appreciate the reviewer comment and agree with it. The description of Hg as toxic element of line 189 has been removed.

Line 195 - the authors should write a definition of “bioavailability”.

We kindly appreciate the reviewer comment and agree with it. It has been written and cited a definition of bioavailability.

Line 203 - Who recommend 1 µg of Hg (II)/L in drinking water?

We kindly appreciate the reviewer comment and agree with it. Thus, a quote has been added.

In the first paragraph there are few quote for a review.

We kindly appreciate the reviewer comment. In this case, a systematic review has been proposed and conducted. Following the standards set by prestigious researchers, such as Codina (Codina, 2015) and Kitchenham et al. (Kinchenham & Charters, 2007), the systematic review has defined the specific scope of the review and the search process conditions (sources, laguages, etc.). As a result, 537 documents have been consulted (210 on arsenic removal, 190 on cadmium and 137 on mercury). From those documents, 31 were classified as reviews. From those selected documents, as it can be seen in the new added table 5, 36 different documments have been used to show the characteristics of the different treatments and processes due to they were the most representative and included the most detailed data. An average revision document includes between 25 and 60 references, thus, 36 documents is a good selection for the revision part of this paper, which not only includes a revision, but also an energy prospects. Moreover, the full paper includes 97 references.

·         Codina, L. (2015). No lo llame Análisis Bibliográfico, llámelo Revisión Sistematizada. Y cómo llevarla a cabo con garantías: Systematized Reviews + SALSA Framework. Retrieved from https://www.lluiscodina.com/revision-sistematica-salsa-framework/

·         Kitchenham, B. A., & Charters, S. (2007). Guidelines for performing Systematic Literature Reviews in Software Engineering. Version 2.3 (EBSE-2007-01). Retrieved from

Figure 3 - You wrote that the period of publication was 2004-2017. Why is 2018 in figure?

We kindly appreciate the reviewer comment and apologize for the inconveniences. The year 2018 has been removed of the figure.

Table 3 - Please, correct Bioleaching from “-e” to “- “.

We kindly appreciate the reviewer comment and apologize for the inconveniences. “e” has been removed of the table.

Line 283 - … Who did determine the reduction of the treatment efficacy?

We kindly appreciate the reviewer comment and agree with it. Thus, a quote has been added.

Line 287 - … And about in situ technology?

Manuscript has been updated with in-situ treatments explanation.

Line 294 - In line you wrote that the optimum pH for As (III) is 7. You have to specify that this pH is optimal in absence of other substances.

We kindly appreciate the reviewer comment and agree with it. Thus, the sentence has been rewritten as: “The optimum pH for As (III) removal is 7, using low intensity, in absence of other substances.”.

Line 310 - Please use an approximately form, for example 67 mg/g

We kindly appreciate the reviewer comment and agree with it. Thus, the digits have been rounded.

Line 311 - mg of As/g adsorbent?

In this case, it has been explained that it refers to “adsorbent”.

Line 320 – Acheampong and Lens […]

We kindly appreciate the reviewer comment and agree with it. Thus, a quote has been added.

Line 336 – What kind of variables? What was optimal number of cycles for As removal?

We kindly appreciate the reviewer comment and agree with it. The required information has been incorporated to manuscript.

Line 339 – Actual normative… in which country?

It has been included that, in this case, the regulations refer to the Netherlands.

Line 351 – Please correct pH value to 3.0; 99.5% after?

We kindly appreciate the reviewer comment and agree with it. The erratum has been corrected. That's right, after the treatment process.

Line 368 – As removal.

We kindly appreciate the reviewer comment and agree with it. It has been included “As removal”.

Line 384 – “Removing cadmium from water is fundamental for human health”. As and Hg removals are not important for human healt? Please delete this sentence.

We kindly appreciate the reviewer comment and apologize for the inconveniences. In this case, the sentence has been removed.

Line 401 – the authors wrote in line 394 that this bioremediation approach decreases the bioavailability (of cadmium?). But if cadmium is less bioavailable, why toxicity will increase? Please, specify.

We kindly appreciate the reviewer comment and agree with it. A new paragraph has been written in order to explain the increase in toxicity.

Line 409 – the authors mean “problem of application in real scale”?

We kindly appreciate the reviewer comment and agree with it. It is clarified that it is real scale.

Line 415 – Please remove “by”.

We kindly appreciate the reviewer comment and apologize for the inconveniences. In this case, the term “by” has been removed.

Line 444 – How much high is the efficiency?

We kindly appreciate the reviewer comment and agree with it. The efficiency values in the process have been added.

Line 459 – as line 310.

We kindly appreciate the reviewer comment and agree with it. Thus, the digits have been rounded.

Line 460 – Please, specify the dose used.

We kindly appreciate the reviewer comment and agree with it. In is specified that Cd solution is 500 ml with the concentration of 10 mg/l and pH 6.

Line 465 – Quote these works.

We kindly appreciate the reviewer comment and agree with it. New references have been included in manuscript.

Line 490 – Why was this soil washing sequence better than others?

Wei et al. results demonstrate that the tested six influencing factors except temperature had a marked effect on the efficiency of heavy metal removal.

Line 499 – Why did you investigate about only 2 treatments for Mercury? For others you cited more works. You have to insert at least another 2 treatments (conventional techniques for example).

We kindly appreciate the reviewer comment and agree with it. Three conventional and two innovative technologies are now reported.

Line 522 – How much is the time of exposure?

The exposure time is already indicated, it is 30 days.

Line 538 – Elimination of?

It is eliminated Hg (II) in both cases.

Line 542 – Adsorption of?

We kindly appreciate the reviewer comment. It is included “adsorption of Hg (II)”.

In the second paragraph, you should insert a table with the discussed works and related results.

We kindly appreciate the reviewer comment and agree with it. In order to help the reader with the discussion of the systematic review, Table 4 has been added summarizing the consulted sources for each treatment. This table helps to easily find the references and get the necessary data for comparison purposes.

Line 551 – Efficient than? You done an economic analysis?

A table (Table 5) with different remediation technologies average costs has been added to manuscript. Explanation has been improved in this issue.

Line 552 – I disagree because for example soil washing increase the remediation waste.

We kindly appreciate the reviewer comment and agree with it. The sentence has been rewritten to make it easier to understand.

Line 565 – How much higher?

If adsorption-co-precipitation rates are observed, the rate of arsenic adsorption is found to be 0.90 mM higher than arsenic-ferrihydrite co-precipitation.

Line 568 – Is there a reason?

We kindly appreciate the reviewer comment and agree with it. It has been included the reason.

Line 606 – How much cost conventional treatments? As line 551. If you don’t, please report a quote.

We kindly appreciate the reviewer comment and agree with it. The information required has been reported in manuscript.

Line 611 – Why could this technique be interesting?

We kindly appreciate the reviewer comment. According to Vena et al., this technique shows a great potential reuse waste into functional nanomaterials.

Line 615:619 – You are discussing about Cadmium, please remove discussion about Pb and Cu.

We kindly appreciate the reviewer comment and agree with it. The discussion about Pb and Cu has been removed.

Line 623 – Please quote the literature. You should insert the equation of kinetic.

We kindly appreciate the reviewer comment and agree with it. Thus, a quote has been added and the equation has been inserted.

Line 629 – What are the predicted values?

There was an error of 11% for non-activated carbon and 7% for activated carbon.

Line 638 - … removal after?

We kindly appreciate the reviewer comment and agree with it. Thus, the sentence has been rewritten as: “Results show a cadmium removal efficiency of 97-98% after the previous treatment.”.

Line 645 – Bioavailability of soils?? Maybe, you want to write “… cadmium bioavailability in soil must be optimized to enhance cadmium removal”.

We kindly appreciate the reviewer comment and agree with it. Thus, we have applied the proposed change.

Line 648 – As line 645, it’s incorrect! The bioavailability is the accessible amount of pollutant (in this case cadmium) to organism.

We kindly appreciate the reviewer comment and agree with it. Thus, we have applied the proposed change.

Line 659 – How much smaller?

We kindly appreciate the reviewer comment and agree with it. The efficiencies has been included.

Line 662 – Remove BOD abbreviation because it’s useless for 1 time.

We kindly appreciate the reviewer comment and agree with it. The term has been removed.

Line 667 – Please correct 7751 to 7,751.

We kindly appreciate the reviewer comment and agree with it. The term has been corrected.

Line 668 – 9.0 g of Hg(III)/mL of? Water?

We kindly appreciate the reviewer comment and agree with it. The erratum has been corrected; 9.0 ± 0.4 mg Hg/l.

In the third paragraph, you have to compare conventional Hg treatments with innovative techniques.

We kindly appreciate the reviewer comment and agree with it. Comparison has been reported and manuscript has been improved.

Line 688:690 – Can the authors include a cost-benefit analysis?

We kindly appreciate the reviewer comment and agree with it. A table with treatment average costs has been included in manuscript and a discussion about costs of different technologies has been also reported.

Line 707:715 – It’s too vague. The authors have to explain better the MFC and MEC treatments, especially the way to produce hydrogen and methane. If it’s possible, you can also include an estimate of the revenue.

We kindly appreciate the reviewer's comment and agree with it. Thus, an extended description of the MFC and MEC treatments, focusing in the fundamentals of the hydrogen and methane production has been provided in lines 891-928 of the track changes version of the manuscript and new references [86-97] have been provided for interested readers.

In the fourth paragraph the authors can’t write about a “real” energy analysis because as you wrote in line 692 “… future research lines must be focused … on their energy requirements”. If you can’t do as I wrote for line 688, energy analysis could be integrated in conclusion at line 723 rather than repeat the result discussed in third paragraph.

We kindly appreciate the reviewer's comment and agree with it. Thus, sections 4 (now called “Future prospects”) and 5 (Conclusions) have been reorganized in such a way the Conclusions are more precise, omit the summary of the results discussed in the third section and includes the energetic approach. Then, section 4 has been rewritten focusing in the novel methods of using MECs and MFCs for toxic pollutants treatments due to their energetic advantages.

Reviewer 3 Report

This MS present a systematic review on As, Cd and Hg remediation using several methods. The language need to be improved along the text body to avoid repetition of words in the same sentence for example. It is too long and several parts may be reduced. Please include references along the text. Do authors have authorization to use images from another papers? Please clarify the aims of study. Please reduce the text in conclusion. Other comments: Lines 22-24 – Please rephrase the sentences to avoid repetition of the word “techniques”. Line 27 - because they are ecological? Line 34 – Please clarify the sentence: “Waste is a material … or operation. Line 48 -  European countries Line 65 - Please clarify the sentence: “Extractive industries practice surface or … Lines 70-73 – Please delete the sentences since it is well known: “Heavy metals are … 72 7 g/cm3. Line 90 – Please explain and rephrase accordingly: “specific metabolic routes? Is it related to the following sentence (line 120) ? (As, Cd, Hg) as these contaminants are non-biodegradable and are not essential for the biomass. Line 228, and 231 – Table 2, and Table 3 – Metal/metalloid Lines 236-242 - Please include references along the text. Please insert Fig 5 after its mention in text. Line 503 – Please use italics for Limnocharis flava

Author Response

REVIEWER 3 COMMENTS

This MS present a systematic review on As, Cd and Hg remediation using several methods. The language needs to be improved along the text body to avoid repetition of words in the same sentence for example. It is too long and several parts may be reduced. Please include references along the text. Do authors have authorization to use images from another papers? Please clarify the aims of study. Please reduce the text in conclusion.

We thank the reviewer for his/her positive appreciation of the proposed manuscript and his/her valuable comments. At this point, according to the reviewer’s remark, the manuscript has been thoroughly revised by an English native-speaker colleague both in grammar and style. Figures and Tables have been also revised and significantly improved and, to our concern, the readability of the manuscript has significantly improved. Figures have been adapted from the original sources and their authorship is now 100% of the manuscript authors.

On the other hand, the reference list has been enlarged until 97 references and the aims of the study have been reinforced: (i) provide a systematic review on the existing treatments for the remediation of hazardous wastes and contaminated soils (focusing in the three most common pollutants: arsenic, cadmium and mercury) and (ii) introduce the need of an energetic analysis of the different treatments to make them more feasible for in situ applications, taking advantage of local energy resources (especially renewable energy resources). The final aim of the authors of the present study is to introduce the state of the art of this topic focusing on the energetic approach in order to present, in future studies, solutions to make these treatments self-sufficient from an energetic point of view.

Finally, it has been done efforts to reduce the text of the manuscript and to include more references to the bibliography, especially in sections 4 and 5. However, some new paragraphs have been added in order to satisfy some editor’s and reviewers’ comments.

Changes between the original submission and the revised version are highlighted in the ManuscriptTrackChanges document. We really want to thank the reviewer for his/her valuable comment which has contributed to increase the overall quality of the manuscript and we expect it now fulfills his/her quality standards.

Other comments:

Lines 22-24 – Please rephrase the sentences to avoid repetition of the word “techniques”.

We kindly appreciate the reviewer comment and agree with it. Thus, the sentence has been rewritten as: “we observed a certain predominance of the use of biotic techniques, compared to a variety of others”.

Line 27 - because they are ecological?

Replaced by “environmentally friendly”.

Line 34 – Please clarify the sentence: “Waste is a material … or operation.

The sentence has been rewritten to make it more understandable.

Line 48 - European countries

We kindly appreciate the reviewer comment and agree with it. Thus, the erratum has been revised.

Line 65 - Please clarify the sentence: “Extractive industries practice surface or …

We kindly appreciate the reviewer comment and agree with it. Thus, the sentence has been rewritten as: “The extractive industry obtains minerals through mining techniques, which include drilling and blasting operations, among others.  In addition, for commercial use, these extracted minerals must benefit from mineral processing techniques. In both stages of the process, waste is generated.”.

Lines 70-73 – Please delete the sentences since it is well known: “Heavy metals are … 72 7 g/cm3.

We kindly appreciate the reviewer comment and agree with it. Thus, the sentence has been removed.

Line 90 – Please explain and rephrase accordingly: “specific metabolic routes? Is it related to the following sentence (line 120)? (As, Cd, Hg) as these contaminants are non-biodegradable and are not essential for the biomass.

We kindly appreciate the reviewer comment and agree with it. Thus, the paragraph has been rewritten.

Line 228, and 231 – Table 2, and Table 3 – Metal/metalloid

We kindly appreciate the reviewer comment and agree with it. Thus, in the header of the tables 2 and 3, the term “metalloid” has been included.

Lines 236-242 - Please include references along the text.

We kindly appreciate the reviewer comment and agree with it. Thus, we have included 5 new quotes along the lines 287-292 in the track changes version of the manuscript.

Please insert Fig 5 after its mention in text.

We kindly appreciate the reviewer comment and apologize for the inconveniences. Effectively, figures must be mentioned in text before their apparition. In this case, we have included references to figure 5 in line 376 in the track changes version of the manuscript.

Line 503 – Please use italics for Limnocharis flava

We kindly appreciate the reviewer comment and agree with it. We have used italics for Limnocharis flava in line 526 in the track changes version of the manuscript.

Round  2

Reviewer 2 Report

Line 522, After line 544 the authors introduced chemical extractions. Please change the title of subsection 2.3.1. Line 526, In situ thermal desorption method was also applied simultaneously heating the soil with resistors (usually 400-600 °C) and using a vacuum for a gas emission. The soil may be heated with a steam high-flow at high temperatures through injection wells. Line 544, Please change “e” with “and”. Line 562, With this temperature, it may be find the presence of mercury in the collected gas.  Line 633, The authors could specify that the excavation and transport costs increase the total amount. Line 712. Please change “k_e” with “k_2”. Line 752, Do the cost include the landfilling of activated carbons? Line 822, Please change “HCO3” to “HCO3-“. Did the authors report if the biomethane could be used to cover the costs of remediation?

Author Response

REVIEWER 2 COMMENTS

We thank the reviewer for his/her positive appreciation of the proposed manuscript and his/her valuable comments. Undoubtedly, they have contributed to increase the overall quality of the manuscript and we expect it now fulfills his/her quality standards.

Line 522, After line 544 the authors introduced chemical extractions. Please change the title of subsection 2.3.1.

We kindly appreciate the reviewer comment and apologize for the inconveniences. Chemical extraction was another method explained in this section (as it was previously presented in the first paragraph of section 2.3), but the title and the first words of the paragraph were missing. Now, a new section: 2.3.2 has been added.

Line 526, In situ thermal desorption method was also applied simultaneously heating the soil with resistors (usually 400-600 °C) and using a vacuum for a gas emission. The soil may be heated with a steam high-flow at high temperatures through injection wells.

We kindly appreciate the reviewer comment and agree with it. Thus, in-situ thermal desorption description has been added to the manuscript.

Line 544, Please change “e” with “and”.

We kindly appreciate the reviewer comment and apologize for the inconveniences. “e” has been changed for “and”.

Line 562, With this temperature, it may be found the presence of mercury in the collected gas.

We kindly appreciate the reviewer comment and agree with it. Thus, a vitrification description has been detailed and the manuscript has been improved.

Line 633, The authors could specify that the excavation and transport costs increase the total amount.

We kindly appreciate the reviewer comment. Ex-situ description has been improved with the reviewer suggestion.

Line 712. Please change “k_e” with “k_2”.

We kindly appreciate the reviewer comment and apologize for the inconveniences. “k_e” has been changed for “k_2”.

Line 752, Do the cost include the landfilling of activated carbons?

We kindly appreciate the reviewer comment. Effectively, landfilling of activated carbon particles is included in the cost breakdown. This information has been added to the manuscript.

Line 822, Please change “HCO3” to “HCO3-“. Did the authors report if the biomethane could be used to cover the costs of remediation?

We kindly appreciate the reviewer comment and apologize for the inconveniences.

 The authors would like to express their sincere gratitude to the reviewer for his/her time and suggestions, as well as for his/her commitment to improve the manuscript. His/her advice and recommendations throughout the review round have been very beneficial and valuable.

 Best regards,

THE AUTHORS.

Reviewer 3 Report

Authors have done major required changes and MS was improved.

Author Response

The authors would like to express their sincere gratitude to the reviewer for his/her time and suggestions, as well as for his/her commitment to improve the manuscript. His/her advice and recommendations throughout the review round have been very beneficial and valuable.

 Best regards,

THE AUTHORS.